# Factors shaping the gender wage gap among college-educated computer science workers

**Sharon Sassler**[1]*, **Pamela Meyerhofer**[2]

**1** Department of Sociology, The Jeb E. Brooks School of Public Policy, Cornell University, Ithaca, New York, United States of America, **2** Federal Trade Commission, Washington, DC, United States of America

\* Sharon.Sassler@cornell.edu

## Abstract

Encouraging women to pursue STEM employment is frequently touted as a means of reducing the gender wage gap. We examine whether the attributes of computer science workers–who account for nearly half of those working in STEM jobs–explain the persistent gender wage gap in computer science, using American Community Survey (ACS) data from 2009 to 2019. Our analysis focuses on working-age respondents between the ages of 22 and 60 who had a college degree and were employed full-time. We use ordinary least squares (OLS) regression of logged wages on observed characteristics, before turning to regression decomposition techniques to estimate what proportion of the gender wage gap would remain if men and women were equally rewarded for the same attributes–such as parenthood or marital status, degree field, or occupation. Women employed in computer science jobs earned about 86.6 cents for every dollar that men earned–a raw gender gap that is smaller than it is for the overall labor force (where it was 82 percent). Controlling for compositional effects (family attributes, degree field and occupation) narrows the gender wage gap, though women continue to earn 9.1 cents per dollar less than their male counterparts. But differential returns to family characteristics and human capital measures account for almost two-thirds of the gender wage gap in computer science jobs. Women working in computer science receive both a marriage and parenthood premium relative to unmarried or childless women, but these are significantly smaller than the bonus that married men and fathers receive over their childless and unmarried peers. Men also receive sizable wage premiums for having STEM degrees in computer science and engineering when they work in computer science jobs, advantages that do not accrue to women. Closing the gender wage gap in computer science requires treating women more like men, not just increasing their representation.

## Introduction

In 2019, women who worked full-time, year-round earned 82 percent of what their male counterparts earned [1]. Progress towards narrowing the gender wage gap has largely stalled since the 1990s. Women are under-represented in employment sectors with above average wages

**Data Availability Statement:** The data underlying the results presented in the study are available from the IPUMS USA database (https://www.ipums.org/) for the American Community Survey.

**Funding:** SS received an award from the National Science Foundation (NSF), entitled "Early Career

Transitions into STEM Employment: Processes Shaping Retention and Satisfaction," supported by the National Science Foundation (Award # 1432522) under the Directorate for Education and Human Resources (EHR) (www.NSF.Gov). PM also received supplementary support from the Cornell Population Center (https://cpc.cornell.edu). The Funders had no role in study design, data collection and analysis, decision to publish, or preparation of the manuscript.

**Competing interests:** The authors have declared that no competing interests exist.

such as STEM (Science, Technology, Engineering, and Mathematics) occupations, where the gender wage gap is smaller than in the overall labor force [2, 3]. Encouraging more women to pursue STEM jobs is often proffered as an important step towards reducing the gender wage gap [4, 5].

But not all STEM occupations are created equal. Computer science jobs are among the STEM occupations with the highest annual earnings [3, 6, 7] and best job prospects. Furthermore, roughly one-half of all STEM jobs are in computer science [7], yet the field is often perceived as being inhospitable to women [8, 9]. Women employed in computer science are more likely to report having experienced gender discrimination at work than women employed in other STEM or non-STEM occupations [10], and their underrepresentation exacerbates the "chilly climate" effect [11]. In 2015, only 26 percent of those working in computer and mathematical science were women [12]–a substantial reduction from the mid-1980s, when women made up 35 percent of all computer science workers [5, 8]. The evidence suggests barriers to increasing women's representation in the field, including maternal profiling, implicit gender biases, and sex typing of occupations [3, 13–15]. Given the importance of technology jobs for national competitiveness, understanding what factors impede the attainment of gender parity in wages among those working in this field is important for addressing barriers to achieving gender equality.

Numerous programs exist encouraging girls and women to embrace computer science. Such ventures urge girls and young women to invest in the training to enable them to obtain jobs in computer science by promoting the pursuit of college majors or advanced degrees in related fields, and address gender differences in human capital. Even when women have the appropriate training, however, they often are valued less than equivalent males for identical positions [16] and are funneled into lower-paying jobs [13]. Family formation–getting married, having children–also has different costs for women and men. Married men and fathers receive wage premiums for their family responsibilities [17], but far more variability is evident in how women who are married or mothers are remunerated in the paid labor force. Studies conducted on workers in the closing years of the 20th century often documented a parenthood wage penalty for mothers [17–19], though research on the wages of professionals in the new millennium suggests that the returns to parenthood are changing. A growing body of evidence finds that both men and women receive a parenthood bonus, at least among those who are highly educated [3, 6, 19, 20]. These findings, however, run counter to the narrative that mothers face especially steep barriers to attaining equality in the paid labor force relative to their male counterparts [21, 22].

Using American Community Survey (ACS) data from 2009 to 2019, and focusing on working-age respondents employed full-time, we examine what factors are associated with the gender wage gap in computer science. If reducing the gender wage gap is mainly a function of compositional effects–getting more women trained in STEM fields or encouraging them to pursue jobs in the most lucrative computer science fields–then accounting for such factors will reduce observed wage disparities. On the other hand, women and men may receive different returns to similar attributes–the classic definition of discrimination. One strength of utilizing ACS data is that it enables us to explore the population of those working in computer science, a considerable advance over STEM research that relies on samples of those who obtained their degrees in computer science or related STEM fields [3, 23–26]. We find evidence of a persistent gender wage gap in computer science. But differential returns to attributes explain far more of the gender wage gap than do compositional effects. In other words, discrimination continues to account for a substantial portion of the gender wage gap in computer science. Until women are rewarded the same as their equivalent male counterparts working in computer science jobs, the gender wage gap and attrition from such jobs will persist [23, 24, 27].

## Understanding the persistent gender wage gap

An extensive body of research explores the stubborn persistence of the gender wage gap, and reasons why progress in narrowing the gap has largely stalled [2]. Traditional explanations for the wage gap have focused heavily on how gendered role expectations for partnering and parenting shape employment and wages [25, 28]. Others emphasize gendered disparities in human capital accumulation, such as choice of college major and work histories [29, 30]. A third theoretical tradition explores demand-side factors, such as employer discrimination against women and minorities, to explain persistent gendered wage inequities [2, 31, 32]. Below we review how such explanations might contribute to the gender wage gap in computer science occupations.

**Family explanations.** Explanations for the gender wage gap often focus on gendered norms that assign primacy to women's obligations as wives and mothers. Numerous studies have documented an earnings bonus associated with marriage, though more so for men than women [18, 33, 34]. But union formation may reduce even childless women's availability for paid employment, as women still perform a larger share of domestic chores than men [35, 36], and the amount of time spent in domestic labor increases significantly when women (but not men) enter cohabiting or marital unions [37]. Recent studies focused on STEM professionals have found that the wage gap among the married is considerably larger than among single workers [6], highlighting the gendered nature of the institution of marriage. The long-term impacts of marriage on earnings may also be found among those who divorced, though previously married women experience more negative economic effects than do men, especially if they have children [38].

Furthermore, the costs of parenting are not born equally by men and women. For women, parental status is often associated with reduced wages, highlighting a "motherhood penalty," whereas among men paternal status offers wage advantages known as "the fatherhood premium" [16, 17, 20]. The largest wage penalties are experienced by the most privileged women, those with skills in high demand and who are the best remunerated, as large returns to labor market experience make even small amounts of time out of employment for childrearing costly [30]. Work interruptions for childbearing and -rearing mean that parents' skills deteriorate during care-related career breaks [39], which may be particularly salient for jobs in computer science, a rapidly changing field that demands constant skill upgrading. As maternal employment becomes normative and the costs of leaving the work force remain high, professional women–even mothers of young children–are more likely than previous cohorts to remain in the work force [18, 39]. Nonetheless, even relying on the best measures of job experience and employment tenure does not eliminate the child penalty for American mothers in the overall labor force [40].

Determining the size of the penalties or premiums for parenthood, and whether they have changed over time for mothers and fathers, remains challenging [17, 34, 40]. Even though recent cohorts of fathers are playing a larger role in childrearing, working women continue to do more of both childcare and domestic labor [41]. This also appears to be the case among women employed in STEM fields, where women are almost twice as likely to reduce their work efforts following childbirth than are men [30]. In fact, in many professions men often double down on being "ideal workers" following the arrival of children, working even longer hours [34, 41]. The fatherhood premium ranges from 3 to 10 percent, depending upon the population considered or model specifications [17, 33–34, 41]; over the past few decades' men's earnings premiums have increased more for highly educated fathers [18, 42]. The evidence suggests a fatherhood premium persists among computer science workers, though it is

unclear how large it would be in a field where those with the newest degrees (and most recently acquired skills) can demand high wages.

There is less consensus about the impact of parenthood for women. The motherhood penalty began decreasing in the 1990s [18, 19], particularly among highly educated women. Glauber [18] found that by the early 2010s the motherhood penalty for high-earning women was eliminated. Others also found that professional mothers are no longer disadvantaged, in terms of their wages, relative to childless women [19, 20]. This also appears to be the case among women with STEM degrees [3, 6], though whether this finding holds for those working in the computer science workforce is an open question.

Confounding the associations between marriage, parenthood, and earnings are age and cohort effects, although these are also often used as stand-ins for labor market experience [3]. Marital delay has helped push back the age of first parenthood among professionals, given that among the college-educated in the United States marriage generally remains a prerequisite for parenthood [43]. Among the college-educated, parenthood is frequently delayed into the third decade [44], thereby increasing the period during which women work and accumulate employment experience prior to family building [27]. The deferral of parenthood into the thirties means that women's early employment trajectory often closely resembles that of men; employees in their twenties also have college degrees of a relatively recent vintage, where near-parity in earnings is more evident than among older cohorts [45].

Studies of earlier cohorts of women have noted that the gender wage gap expands over the life course, though whether this pattern remains evident among recent cohorts is understudied. Focused on a cohort of British women born in 1958, Joshi and colleagues found that the initial gender wage gap among young workers widened substantially during the childbearing years, largely due to family formation that affected work experience [46]. This finding also emerges in more recent samples of American professional workers, including those examining STEM professionals [3, 6], though the gap is field specific and smaller in magnitude among more recent cohorts. Nonetheless, scholars point to the expansion of the gender wage gap over the course of women's careers in computer science as evidence of a glass-ceiling effect [3], where discriminatory barriers experienced throughout the career prevent women from advancing to positions of authority simply because they are women [47].

**Human capital explanations.** Economic explanations for the persistent wage gap typically focus on differential human capital investments to explain why women continue to earn less than men. Gender variation in fields of study, differences in hours worked, returns to job experience and educational attainment, and job sector are used to explain wage disparities. According to this perspective, women are less likely to pursue college majors that lead to demanding and time-intensive occupations than men, eschewing them for majors that are perceived as leading to more flexible (e.g., family friendly) occupations [28, 48]. In recent decades, women have increasingly pursued STEM degrees, and young women no longer plan their occupational goals around caregiving or family plans [49, 50]. Yet differences in types of degrees obtained persist, and women are considerably less likely to major in the STEM fields with the highest returns–engineering and computer science–than men, which contributes to the persistence of the gender wage gap [3].

Furthermore, gender disparities in human capital accumulation persist. Once in the work force, women work fewer hours than men, on average, and are less likely to work continuously [2, 18, 41], resulting in less work experience [50, 51], which affects pay and promotions. Yet human capital explanations have become weaker predictors of wage discrepancies. For starters, women have surpassed men in college attendance and receipt of Bachelors' and Masters' degrees [50]. Furthermore, women's likelihood of remaining in the labor force following parenthood increased in the latter half of the 20th century, before stalling in the new century [19,

52, 53]. More recent studies confirm that accounting for experience reduces much of the earnings penalty experienced by women, particularly those who were mothers [17, 40]. Among recent college graduates, formal hiring practices, diversity initiatives, parental leave policies and selection into motherhood may lead to greater wage parity [19, 20], though the presence of older workers continues to shape returns experienced by women overall, contributing to the gender wage gap.

As human capital's contribution to the gender wage gap has decreased, the importance of industry and occupation have grown. In 1980, gender gaps in industry and occupation accounted for 20 percent of gender pay disparities, but by 2010 they made up 51 percent of the gender gap–although the wage gap had shrunk over time [2]. Women in STEM are less likely to work in jobs perceived as more 'masculine' [13, 14]; these are also occupations that pay more, and where pay and prestige declines as their share of female employees increases [3, 54]. Others have found that the greatest economic rewards among technology workers accrue to those engaged in programming-intensive roles–which have the lowest shares of women; Cheng and colleagues [55] found that men experienced greater employment growth in programming-intensive occupations than women, contributing to the stalled convergence of the gender wage gap among college graduates. But researchers have noted gender wage disparities among those working in similar scientific occupations [54]. Slow progress in reducing within-occupation sex segregation perpetuates gender pay disparities [2, 42, 56].

**Discrimination.** A third potential explanation for the gender wage gap centers on employer discrimination, where women are perceived as less dedicated or career focused than their male counterparts. Hegemonic gender beliefs undergird employers' views of men's and women's abilities, and shape how women are evaluated and remunerated [57]. Even though discrimination's contribution to the gender wage gap in the overall labor market decreased between 1970 and 2010 [32], it continues to influence earnings in STEM fields. Audit studies have shown that employers would pay women entering the STEM labor force less than men with identical characteristics [15, 58]. There is also evidence that women are sorted into lower-level job queues than men when applying for jobs at small- and medium-sized high-tech firms [13]. Women working in computer science jobs also appear to benefit less from experience and human capital accumulation than similar men; researchers using the National Science Foundation's Scientists and Engineers Statistical Data System (SESTAT) from 1995 through 2008 to examine wages of women with college degrees in computer science who worked in related occupations found little evidence of a cohort change in the gender wage gap [3].

Discrimination may also operate through maternal profiling, which is exacerbated in male-dominant fields. Married women are often perceived as less committed workers [21, 59, 60], but young single women are also assessed as being less competent and worthy of raises [16]. Despite demonstrating equivalent or greater work effort, work intensity, and job engagement, mothers were assessed as being less dedicated workers than their male counterparts who were fathers [59]. Even the possibility that women may become mothers serves to disadvantage women in demanding professions [60]. Social scientists examine the extent to which the labor market characteristics associated with productivity (experience, education) are rewarded differently for men and women, using this to signify discrimination. There may, however, be unobservable differences between men and women that are not captured, which account for differential returns.

**Aims.** The literature reviewed leads us to expect that the previously observed associations between family characteristics, human capital measures, and discrimination will contribute to the gender wage gap among those working in computer science jobs. Whereas accounting for family factors and measures of human capital should reduce the size of the gender wage gap, evidence on how discrimination operates to disadvantage women leads us to anticipate that a

sizable proportion of the gender wage gap will be due to differential returns to men's and women's attributes, particularly those linked to family characteristics. Our analysis proceeds as follows. We first examine descriptive evidence of gender disparities in wages in computer science occupations, highlighting variation by job type. Next, we turn to results of multivariate analysis, using ordinary least squares (OLS) regressions of logged wages on observed characteristics. Third, we employ regression decomposition techniques to estimate what proportion of the gender wage gap would remain if men and women had similar characteristics. Last, we present tables and figures demonstrating the differential returns women and men receive for family measures to illustrate the findings from the regression decomposition.

## Data and sample selection

Data are from the American Community Survey, an annual cross-sectional household survey that replaced the long form of the Census in 2000 to provide nationally representative, large sample data between Census years. We use data from the 2009 through 2019 waves of the ACS; 2009 was the year that the ACS began to enquire about the college major for respondents obtaining a bachelor's degree. We end our sample in 2019 to exclude any impact of the 2020 COVID-19 pandemic, shutdown, and recovery. Respondents appear only once in our sample. We limit our sample to individuals working at least thirty-five hours per week who had positive income, were in the prime working age population (age 22 to 60), whose youngest child living in the household is less than 18, and who have (at least) a bachelor's degree. Including part-time STEM workers yielded similar results to models using only full-time workers [2, 3, 42], as only 4.2% of Computer Science workers in our sample are employed part-time (7.6% of women, and 3% of men). The majority of those working in computer science (about 70 percent) have at least a college degree [61], and those with STEM degrees generally find employment soon after degree completion [62, 63]. We exclude those who earned their college degree before 1980, as few individuals working in modern computing jobs earned a degree in computer science before then.

Our most important restriction is that each respondent must be employed in a computer science occupation. The U.S. Census Bureau utilizes SOC (Standard Occupational Classification) codes to categorize computer science occupations [61] and defines the following occupations as computer science: Computer and Information Science (CIS) Manager, Computer Scientist, Computer Analyst, Information Analyst, Computer Programmer, Software Developer, Web Developer, Computer Specialist, Data Administrator, Network Administrator, Network Architect, and All Other Computer Occupations (15–1199 occsoc code). Until 2009, the ACS utilized the 1998 Standard Occupational Classification System. This was updated for 2010 onward with more recent SOC codes. Our initial year of data relies on the 1998 SOC system, with all subsequent years utilizing the 2010 SOC code. Everyone in our sample works in one of these 12 occupations. We only observe those working in a computer science occupation at the time of interview, which means we cannot address differential attrition in computer science by gender prior to observing respondents. Women are substantially more likely than men to leave jobs in computer science [24], but those who remain may be more highly qualified or in better remunerated positions. Consequently, our estimates are likely a lower bound if those most dissatisfied had left prior to being observed. Even after these limitations to the sample, we have 206,640 total observations– 54,510 women and 152,136 men.

**Measurement.** Our *dependent variable* is the logged hourly wages for individuals working in computer science. The ACS provides information on annual earnings from the main occupation, as well as usual weekly hours spent at the main job in the last year. Using these variables, we construct an hourly wage by dividing annual gross salary by 52 (weeks in the year) and by usual hours worked per week. All wages are converted to year 2010 dollars using the

consumer price index. As is customary, we top-code hourly wages at $200 per hour and calculate the log of the wage. Our key *independent variable* of interest is the gender of the respondent, coded as a binary variable (Female = 1; Male = 0).

Our measures of *family characteristics* include respondent's marital and parental status. A dummy variable denotes respondents who are currently married (married = 1). Those that are divorced, separated, or widowed are measured with a dummy variable for previously married, while those who are never married serve as our reference group. Parental status is measured via two ACS questions on number of children in the household (number of children under 18 in the household and number of children under age 5 in the household), which enables us to create four mutually exclusive groups; childless respondents serve as the reference category. Reports of the number of children under the age of five in the household allowed the construction of a dummy variable denoting the presence of only pre-school age (age 0–4) children (Child < 5 = 1). A second dummy variable equals one if all the children in the household are school-aged (ages 5 through 18). Our third dummy variable captures if there was at least one pre-school aged child as well as at least one school-aged child. Unlike the first two measures, this variable requires at least two children under 18 in the household. This measure allows us to ascertain the demands children of different ages impose on parents' ability to work, though it captures only children that live in the household; we do not observe children over the age of 18 or minors residing elsewhere (such as with a different parent). Our measures therefore understate parental obligations, more so for men than women.

We control for age and its square to account for its non-linear impact over the work life course. Unfortunately, the ACS does not ascertain the year of degree receipt, nor does it include a measure of years spent in the labor force, at a particular job, or with a specific firm. An additional measure approximating year of degree receipt was created by adding 22 to the year of birth, but this measure is strongly correlated with our age measure; we therefore use it only for descriptive purposes.

Our measures of *human capital accumulation* include the field in which respondents earned their college degree, level of educational attainment, and specific occupation in computer science. Computer science serves as the reference group for college major, with dummy variables denoting other majors prevalent among those working in computer science occupations–Engineering, Other STEM majors, Business, and a catchall we designate as other non-STEM degree; the five most common majors in this group, accounting for nearly 60% of this category, are social sciences, communications, fine arts, psychology, and English language, literature, and composition. We also include an indicator for having a master's degree, a doctorate, or a professional degree, relative to a bachelor's degree (the reference); unfortunately, the ACS does not collect information on the field of study of the graduate degree. Our final indicator of human capital differentiates between twelve computer science occupations, with software developers serving as the omitted group.

Last, we include controls for race-ethnicity, where we distinguish between non-Hispanic White (reference group), non-Hispanic Black, non-Hispanic Asian, non-Hispanic other race, and Hispanic. We also include a dummy variable indicating whether respondents were born outside the United States to noncitizen parents (foreign-born = 1; native-born = 0). We do not control for region or year as these controls add little explanatory power but cost a large increase in the degrees of freedom (results not shown).

## Who works in computer science?

**Descriptive results.**   Less than a third of workers in occupations classified as "computer science"–only 29 percent–have a college degree in computer science. But combined with those

who obtained degrees in other STEM fields, such as Engineering or the Life and Physical Sciences, STEM degree holders accounted for over half (55 percent) of those working in computer science. Nearly a fifth of those working in computer science jobs obtained their college degree in business, while a quarter have degrees in other non-STEM fields, including a large representation of those with degrees in the social sciences or humanities. There is considerable variability in the proportion of workers with each degree across computer science occupations, as shown in Fig 1. The occupations with the largest share of workers with computer science degrees are computer programmers (41 percent), information analysts (36 percent), and software developers (36 percent). Sizable proportions of software developers (30 percent) also obtained college degrees in engineering. Computer analysts and CIS managers have lower share of workers with college degrees in computer science (18 percent and 23 percent, respectively), and CIS Managers have larger proportions with degrees in business (24 percent) or other non-STEM disciplines (26 percent). Further details on the composition of each computer science occupation–the proportion of women working in each, as well as mean wages and mean usual hours of work, are presented in S1 File.

We present means and standard deviations for the entire sample, as well as separately for women and men, in Table 1. The final columns show the difference between men and women and the p-value of a t-test. As expected, only a quarter of computer science workers (26

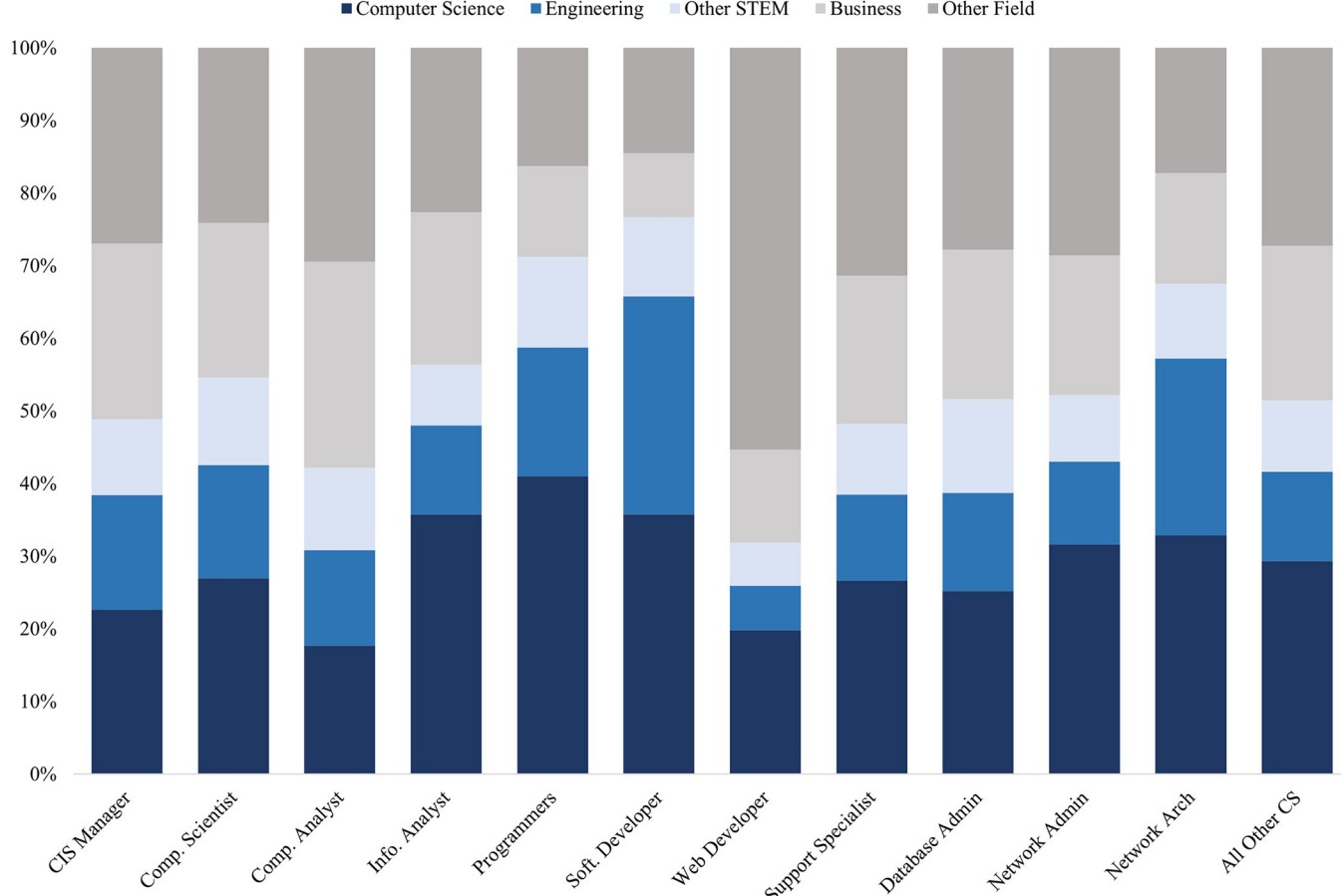

**Fig 1. Top degree fields by occupation.** Notes: Data come from the 2009–2019 American Community survey. The sample is restricted to those ages 22 to 60 with a Bachelor's degree working full time (35+ hours/week) with positive income in a computer science occupation (defined by the Census Bureau in Landivar [61]) whose youngest residential child is under 18. Respondents must also have a valid birthyear and received their degree after 1980. Other STEM degree includes Engineering Technologies, Biology and Life Sciences, Math and Statistics, and Physical Sciences. All remaining majors are in Other non-STEM.

percent) are female. Furthermore, women employed in computer science jobs earned significantly less than their male counterparts, with an average hourly wage of $33.03 versus men's hourly wage of $38.50. That is, women earned about 86 cents for every dollar that men earn. This raw gender wage gap is smaller in computer science than it is for the overall labor force (where the wage gap is 18 percent, with women earning 82 percent of what their male counterparts earn) [1]. Women also work, on average, about an hour less per week than men.

Among men and women working in computer science jobs, we observe considerable gender disparities in family roles that may influence availability for work. About two-thirds of men working full-time in computer science, for example, are married (66 percent), compared with only 58 percent of women. Women whose jobs are in computer science are substantially more likely than their male counterparts to be divorced (12 percent versus 6 percent). Furthermore, men are more likely to have parental obligations than their female counterparts; one-in-five men working in computer science live with preschool aged children, compared with just 15 percent of women, though similar shares reside only with school-aged children (28 percent).

The descriptive results reveal no significant age difference between men and women working in computer science occupations. Additional exploration of the age distribution of the men and women in our sample, however, reveals that women are significantly more likely to be recent graduates or older, whereas men are more likely to be concentrated in the middle ages where family responsibilities are more prevalent. Another important way in which women working in computer science differ from their male counterparts is their racial diversity, though Blacks and Hispanics are less well represented in the computer science workforce than they are in the population overall. Regardless of sex, almost a quarter of those working in the computer science labor force are Asian, and nearly one- third are foreign-born.

Women working in computer science are significantly less likely than their male counterparts to have obtained their degree in computer science (22 percent versus 32 percent) and engineering (9 percent versus 19 percent) (p < .001). On the other hand, a greater proportion of women have degrees in business (23 percent versus 18 percent) or in another non-STEM field (36 percent versus 21 percent), and women working in computer science are more likely than men to have obtained a master's degree. Gender disparities in job specialization also emerge. Women are significantly more likely to work as CIS managers (19 percent) or computer analysts (20 percent) than men (17 percent and 13 percent, respectively), whereas more men work in programming-intensive jobs as software developers (15 percent versus 11 percent of women) or network architects (3 percent versus 1 percent)–jobs with among the highest average wages.

The variation in hourly wages by kind of computer science job held and by sex are shown in Fig 2. Consistent with the descriptive results of Table 1, men earn statistically significantly more than women in all computer science jobs, though there is considerable variability in hourly wages across jobs. Those working as CIS managers earn the most, followed closely by software developers and network architects. Web developers and computer support specialists exhibit the lowest hourly wages. The gap between the wages of men and women is considerably greater for those working as CIS managers and software developers, two of the most common computer science occupations, which account for over thirty percent of women and men working in computer science occupations. The wage gap is often smaller in other occupations, but those jobs also pay less.

## Multivariate results

We turn now to our multivariate results to assess what factors narrow or exacerbate the gender wage gap in computer science. Table 2 shows regression results pooling men and women.

**Table 1. Summary statistics.**

| | | (1) | | (2) | | (3) | | (4) | |
|---|---|---|---|---|---|---|---|---|---|
| | | Full Sample | | Female | | Male | | Difference (2)—(3) | |
| | | Mean | SD | Mean | SD | Mean | SD | Beta | p-value |
| **Key Variables of Interest** | | | | | | | | | |
| | Female | 0.26 | 0.44 | 1.00 | 0.00 | 0.00 | 0.00 | 1.00 | . |
| | Hourly Wage | 37.08 | 21.94 | 33.03 | 18.50 | 38.50 | 22.86 | -5.81 | 0.00 |
| | Annual Income in Thous. $ | 84.27 | 56.65 | 73.74 | 45.91 | 87.97 | 59.53 | -15.23 | 0.00 |
| | Usual Hours/Week | 43.34 | 6.52 | 42.75 | 5.97 | 43.54 | 6.69 | -0.82 | 0.00 |
| **FAMILY CONTROLS** | | | | | | | | | |
| | Married | 0.63 | 0.48 | 0.58 | 0.49 | 0.66 | 0.48 | -0.08 | 0.00 |
| | Never Married | 0.29 | 0.45 | 0.31 | 0.46 | 0.28 | 0.45 | 0.03 | 0.00 |
| | Prev. Married | 0.08 | 0.26 | 0.12 | 0.32 | 0.06 | 0.24 | 0.05 | 0.00 |
| | No Children | 0.53 | 0.50 | 0.57 | 0.50 | 0.52 | 0.50 | 0.06 | 0.00 |
| | Only Child(ren) 0–5 | 0.11 | 0.31 | 0.09 | 0.29 | 0.11 | 0.32 | -0.02 | 0.00 |
| | Young & School-age Children | 0.08 | 0.27 | 0.06 | 0.24 | 0.09 | 0.28 | -0.03 | 0.00 |
| | Only Child(ren) 5–18 | 0.28 | 0.45 | 0.28 | 0.45 | 0.28 | 0.45 | -0.01 | 0.00 |
| **HUMAN CAPITAL CONTROLS** | | | | | | | | | |
| | Age | 38.74 | 9.15 | 38.77 | 9.41 | 38.72 | 9.06 | 0.07 | 0.14 |
| **Degree** | | | | | | | | | |
| | Bachelor's in CS | 0.29 | 0.45 | 0.22 | 0.41 | 0.32 | 0.47 | -0.10 | 0.00 |
| | Bachelor's in Engineering | 0.16 | 0.37 | 0.09 | 0.29 | 0.19 | 0.39 | -0.09 | 0.00 |
| | Bachelor's in Other STEM field | 0.10 | 0.31 | 0.11 | 0.31 | 0.10 | 0.30 | 0.00 | 0.01 |
| | Bachelor's in Business | 0.19 | 0.39 | 0.23 | 0.42 | 0.18 | 0.38 | 0.05 | 0.00 |
| | Bachelor's in Other non-STEM | 0.25 | 0.43 | 0.36 | 0.48 | 0.21 | 0.41 | 0.14 | 0.00 |
| | Professional Degree | 0.01 | 0.11 | 0.01 | 0.11 | 0.01 | 0.10 | 0.00 | 0.03 |
| | Masters Degree | 0.28 | 0.45 | 0.31 | 0.46 | 0.27 | 0.44 | 0.04 | 0.00 |
| | Doctorate | 0.02 | 0.13 | 0.01 | 0.12 | 0.02 | 0.14 | -0.01 | 0.00 |
| **Occupation** | | | | | | | | | |
| | CIS Manager | 0.17 | 0.38 | 0.19 | 0.39 | 0.17 | 0.37 | 0.02 | 0.00 |
| | Computer Scientist | 0.02 | 0.15 | 0.02 | 0.15 | 0.02 | 0.15 | 0.00 | 0.01 |
| | Computer Analyst | 0.15 | 0.35 | 0.20 | 0.40 | 0.13 | 0.33 | 0.08 | 0.00 |
| | Information Security Analyst | 0.02 | 0.14 | 0.01 | 0.12 | 0.02 | 0.15 | -0.01 | 0.00 |
| | Computer Programmers | 0.12 | 0.33 | 0.10 | 0.31 | 0.13 | 0.34 | -0.03 | 0.00 |
| | Software Developer | 0.14 | 0.35 | 0.11 | 0.31 | 0.15 | 0.36 | -0.05 | 0.00 |
| | Web Developer | 0.04 | 0.20 | 0.06 | 0.24 | 0.04 | 0.19 | 0.02 | 0.00 |
| | Computer Support Specialist | 0.11 | 0.31 | 0.10 | 0.31 | 0.11 | 0.31 | 0.00 | 0.11 |
| | Database Administrator | 0.03 | 0.18 | 0.04 | 0.20 | 0.03 | 0.17 | 0.01 | 0.00 |
| | Network Administrator | 0.06 | 0.23 | 0.04 | 0.20 | 0.06 | 0.24 | -0.02 | 0.00 |
| | Network Architect | 0.02 | 0.15 | 0.01 | 0.10 | 0.03 | 0.17 | -0.02 | 0.00 |
| | All Other Computer Science | 0.11 | 0.31 | 0.10 | 0.30 | 0.11 | 0.32 | -0.01 | 0.00 |
| **DEMOGRAPHIC CONTROLS** | | | | | | | | | |
| **Race/Ethnicity** | | | | | | | | | |
| | White | 0.61 | 0.49 | 0.57 | 0.50 | 0.62 | 0.49 | -0.05 | 0.00 |
| | Black | 0.07 | 0.25 | 0.10 | 0.30 | 0.06 | 0.24 | 0.03 | 0.00 |
| | Asian | 0.24 | 0.43 | 0.25 | 0.43 | 0.23 | 0.42 | 0.02 | 0.00 |
| | Other Race | 0.03 | 0.16 | 0.03 | 0.16 | 0.03 | 0.16 | 0.00 | 0.01 |
| | Hispanic | 0.06 | 0.24 | 0.06 | 0.23 | 0.06 | 0.24 | 0.00 | 0.32 |
| | Foreign Born | 0.31 | 0.46 | 0.30 | 0.46 | 0.31 | 0.46 | 0.00 | 0.86 |

*(Continued)*

**Table 1.** (Continued)

|  |  | (1) | | (2) | | (3) | | (4) | |
| --- | --- | --- | --- | --- | --- | --- | --- | --- | --- |
|  |  | Full Sample | | Female | | Male | | Difference (2)—(3) | |
|  |  | Mean | SD | Mean | SD | Mean | SD | Beta | p-value |
|  | Observations | 206,646 | | 54,510 | | 152,136 | | 206,646 | |

Notes: Data come from the 2009–2019 American Community survey. The sample is restricted to those ages 22 to 60 with a Bachelor's degree working full time (35 + hours/week) with positive income in a computer science occupation. A computer science occupation is defined by the Census Bureau in Landivar [61]. We additionally exclude those whose youngest residential child is over 18. Respondents must also have a valid birthyear and received their degree after 1980. Averages are weighted using the ACS person weights.

Model 1 includes an indicator for female, representing the overall raw male-female wage gap in computer science occupations. Model 2 adds in family characteristics and our measures of age, while Model 3 incorporates the human capital measures of college degree field, highest degree, and the 12 computer science occupations. Running separate models with just college

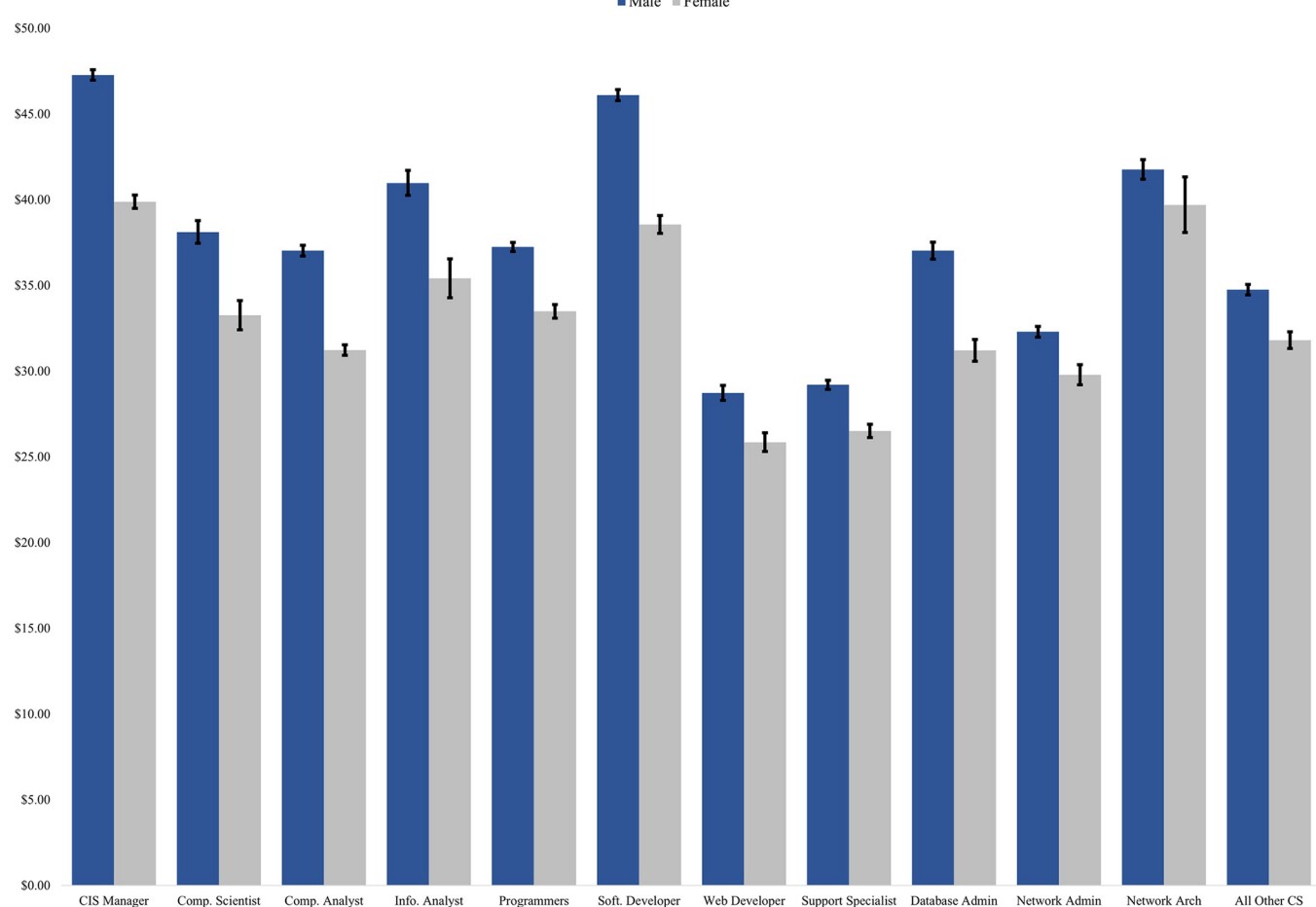

**Fig 2. Average wages by sex and occupation.** Notes: Data come from the 2009–2019 American Community survey. The sample is restricted to those ages 22 to 60 with a Bachelor's degree working full time (35+ hours/week) with positive income in a computer science occupation (defined by the Census Bureau in Landivar [61]) whose youngest residential child is under 18. Respondents must also have a valid birthyear and received their degree after 1980. Averages are weighted using the ACS person weights. Black bars show 95 percent confidence interval.

degree field and educational attainment, or with all education variables and occupations, did not further reduce the gender wage gap, so for the sake of model parsimony we focus on the more inclusive model. Last, in Model 4 we include controls for race, ethnicity, and nativity.

In Model 1, with no other controls, we estimate an overall male-female gender wage gap of 0.144 log points. Hourly wages earned by women who work in computer science are about 13.4 percent lower than the hourly wages earned by men (to estimate the percent from the coefficient, use the following formula: $((e^{coefficient}) - 1) * 100$). The inclusion of family characteristics and age measures improves the model fit substantially (Model 2), and the gender wage gap narrows to 0.125 log points, or a difference of 11.8 percent. The results also indicate that being currently married is associated with a sizable wage premium relative to the never married, though we see no significant returns to being previously married. Married computer science professionals earn 13.5 percent more than their never married counterparts. We also observe a sizable wage premium for parenthood, though it varies by age of child and is smallest when all children are school-aged. The effects of age are curvilinear.

The gender wage gap narrows dramatically upon accounting for human capital experience and occupation (Model 3), and the model fit improves significantly. Adding in our final set of controls for race, ethnicity, and nativity changes the gender wage gap minimally, and the inclusion of these controls does not add significantly to the model fit. Furthermore, coefficient size remains largely consistent across Models 3 and 4. Including controls for age, degree field and level, occupation, and race-ethnicity-nativity reduces the gender wage gap 34 percent, to 0.095 log points, or about 9 cents per dollar less per hour. In the final model, the returns to the family measures are somewhat smaller, with the largest reduction in returns to marriage, perhaps reflecting the association between college major choice, marriage, and fertility [64]. Those who are married receive a marriage premium of about 8.2 cents more per hour over their unmarried counterparts, though the previously married do not receive a marriage premium for having been wed in the past. Parents with only pre-school aged children receive the largest premium, of about 6.2 percent, followed by those with both pre-school aged children and school-aged children, who receive a 5.9 percent bonus, relative to their childless counterparts. Those with only school-aged children also earn more than their childless counterparts, but it is the smallest difference, only 3.4 percent.

We see in Table 2 that with increasing age, earnings increase. What effect does this have on the gender wage gap across age groups? We noted above that we cannot measure cohort, but we can study the gender wage gap by age group to observe how it evolves over the career life course. Graphing the gender wage gap by age (Fig 3), showing both the raw gender wage gap (from Model 1) and after controls (Model 4) reveals that the youngest women, those with newly minted college degrees, earn 10 percent more than men–a substantial amount. While average earnings are greater among those in their late 20s than their counterparts who are younger ($26.51 for 25-29-year-olds vs. $18.73 for 22-24-year-olds), the gender wage gap has already emerged among those 25 to 29, though it remains small; women earn 5.4 percent less than men in this age range. As they enter the prime family formation years, however, the gender wage gap increases. It is largest in the early 40s, with female computer science professionals earning 11.7 percent lower hourly wages than men in this age range; the gender wage gap narrows in the fifties. This inverse U-shape wage gap pattern is consistent with other studies following workers over their life course [46]. Additional research is necessary to ascertain if there has been cohort change in the wage gap at similar ages, though that would require panel data with information on earnings across the life course for several cohorts.

Regression results in Table 2 highlight sizable disparities in returns to degree fields among those working in computer science occupations. Those with a bachelor's degree in engineering–disproportionately men–earned 5 percent higher hourly wages than those with bachelor's

**Table 2. Linear regression predicting log hourly wages.**

|  |  | (1) | (2) | (3) | (4) |
|---|---|---|---|---|---|
|  |  | Model 1 ln(wages) | Model 2 ln(wages) | Model 3 ln(wages) | Model 4 ln(wages) |
| **KEY VARIABLE OF INTEREST** |  |  |  |  |  |
|  | Female | -0.144*** | -0.125*** | -0.097*** | -0.095*** |
|  |  | (0.004) | (0.004) | (0.004) | (0.004) |
| **FAMILY CONTROLS** (Ref = Nvr Marr, Childless) |  |  |  |  |  |
|  | Married |  | 0.127*** | 0.090*** | 0.079*** |
|  |  |  | (0.005) | (0.005) | (0.005) |
|  | Prev. Married |  | 0.005 | -0.002 | -0.001 |
|  |  |  | (0.007) | (0.007) | (0.007) |
|  | Only Child(ren) 0–4 |  | 0.070*** | 0.059*** | 0.060*** |
|  |  |  | (0.006) | (0.006) | (0.006) |
|  | Young & School-age Children |  | 0.067*** | 0.054*** | 0.057*** |
|  |  |  | (0.006) | (0.006) | (0.006) |
|  | Only Child(ren) 5–18 |  | 0.042*** | 0.032*** | 0.033*** |
|  |  |  | (0.004) | (0.004) | (0.004) |
| **Age** (Ref = = 22) |  |  |  |  |  |
|  | Age |  | 0.104*** | 0.103*** | 0.104*** |
|  |  |  | (0.002) | (0.002) | (0.002) |
|  | Age Squared |  | -0.001*** | -0.001*** | -0.001*** |
|  |  |  | (0.000) | (0.000) | (0.000) |
| **HUMAN CAPITAL VARIABLES** |  |  |  |  |  |
| **Degree field** (Ref = Computer Science) |  |  |  |  |  |
|  | Bachelor's in Engineering |  |  | 0.057*** | 0.049*** |
|  |  |  |  | (0.005) | (0.005) |
|  | Bachelor's in Other STEM field |  |  | -0.023*** | -0.030*** |
|  |  |  |  | (0.006) | (0.006) |
|  | Bachelor's in Business |  |  | -0.052*** | -0.051*** |
|  |  |  |  | (0.005) | (0.005) |
|  | Bachelor's in Other non-STEM field |  |  | -0.120*** | -0.121*** |
|  |  |  |  | (0.004) | (0.004) |
| **Educational Attainment** (Ref = Bachelor's) |  |  |  |  |  |
|  | Professional Degree |  |  | 0.037** | 0.037** |
|  |  |  |  | (0.016) | (0.016) |
|  | Masters Degree |  |  | 0.098*** | 0.095*** |
|  |  |  |  | (0.004) | (0.004) |
|  | Doctorate |  |  | 0.215*** | 0.209*** |
|  |  |  |  | (0.011) | (0.011) |
| **Occupation** (Ref = Software Developer) |  |  |  |  |  |
|  | CIS Manager |  |  | -0.061*** | -0.055*** |
|  |  |  |  | (0.006) | (0.006) |
|  | Computer Scientist |  |  | -0.193*** | -0.187*** |
|  |  |  |  | (0.011) | (0.011) |
|  | Computer Analyst |  |  | -0.204*** | -0.197*** |
|  |  |  |  | (0.006) | (0.006) |
|  | Information Security Analyst |  |  | -0.110*** | -0.093*** |
|  |  |  |  | (0.012) | (0.012) |

(*Continued*)

**Table 2.** (Continued)

| | | (1) | (2) | (3) | (4) |
|---|---|---|---|---|---|
| | | **Model 1** <br> **ln(wages)** | **Model 2** <br> **ln(wages)** | **Model 3** <br> **ln(wages)** | **Model 4** <br> **ln(wages)** |
| | Computer Programmers | | | -0.217*** | -0.215*** |
| | | | | (0.006) | (0.006) |
| | Web Developer | | | -0.355*** | -0.350*** |
| | | | | (0.010) | (0.010) |
| | Computer Support Specialist | | | -0.426*** | -0.414*** |
| | | | | (0.007) | (0.007) |
| | Database Administrator | | | -0.237*** | -0.233*** |
| | | | | (0.009) | (0.009) |
| | Network Administrator | | | -0.298*** | -0.288*** |
| | | | | (0.007) | (0.007) |
| | Network Architect | | | -0.124*** | -0.113*** |
| | | | | (0.010) | (0.010) |
| | All Other Computer Science | | | -0.291*** | -0.281*** |
| | | | | (0.007) | (0.007) |
| **DEMOGRAPHIC CONTROLS** | | | | | |
| **Race/Ethnicity** (Ref = NH White, Native) | | | | | |
| | Black | | | | -0.123*** |
| | | | | | (0.007) |
| | Asian | | | | 0.029*** |
| | | | | | (0.006) |
| | Other Race | | | | -0.027** |
| | | | | | (0.011) |
| | Hispanic | | | | -0.081*** |
| | | | | | (0.007) |
| | Foreign Born | | | | -0.006 |
| | | | | | (0.005) |
| | Constant | 3.480*** | 1.007*** | 1.255*** | 1.241*** |
| | | (0.002) | (0.037) | (0.036) | (0.036) |
| | Observations | 206,640 | 206,640 | 206,640 | 206,640 |
| | AIC | 403,528 | 369,333 | 352,630 | 351,676 |
| | BIC | 403,549 | 369,425 | 352,906 | 352,003 |
| | R-squared | 0.010 | 0.161 | 0.226 | 0.230 |

Notes: Data come from the 2009–2019 American Community Survey. The sample is restricted to those ages 22 to 60 with a Bachelor's degree working full time (35 + hours/week) with positive income in a computer science occupation. A computer science occupation is defined by the Census Bureau in Landivar [61]. We additionally exclude those whose youngest residential child is over 18. Respondents must also have a valid birthyear and received their degree after 1980. All models are weighted using ACS person weights. Robust standard errors are shown in parentheses. AIC = Akaike information criterion. BIC = Bayesian information criterion.

*** $p < 0.001$

** $p < 0.01$

* $p < 0.05$.

degrees in computer science, while those with college degrees in other STEM fields earned hourly wages that were 3 percent less than their counterparts whose college degrees were in computer science. This earnings gap is, however, smaller than among those who obtained a college degree in business or in another non-STEM major; they earned 5 percent and 11.4 percent less hourly, respectively, than their coworkers with degrees in computer science.

Obtaining additional schooling elevated earnings; the rare computer science professionals with a PhD, for example, earned, on average, 23.2 percent higher hourly wages than their counterparts with only a bachelor's degree, while those with a Master's degree earned 10 percent more.

Wages also differ across jobs within computer science. Software developers (the omitted category) earn significantly more than all other computer science jobs. Computer & Information Systems Managers have the second highest earnings, though they make 5.4 percent less than software developers, followed by Information Security Analysts with an 8.9 cent hourly wage gap. Computer support specialists and web developers exhibit the largest wage gap of all the computer science occupations, with hourly earnings that are 34 percent and 30 percent lower than their counterparts working as software developers.

The results also indicate that Black, Other Race, and Hispanic workers earn significantly less than White workers, while Asian workers receive a premium. Compared to their White counterparts, Black computer science professionals earn 11.6 cents less per hour, while Hispanic computer science workers earn 7.8 percent lower hourly wages. The effect of being foreign-born is not statistically significant, though additional results (not shown) reveal that the Asian earnings advantage is entirely driven by foreign-born workers.

**Accounting for gender differences in returns to attributes.** Our next analysis explores what the gender wage gap in computer science would be if women were rewarded like men for their attributes. We rely on Oaxaca-Blinder regression decomposition techniques to estimate to what extent wage differentials are the result of differences in men's and women's characteristics (compositional effects), and how much of the gender wage gap is due to women receiving different returns to their characteristics than their male counterparts with those attributes (commonly referred to as possible discrimination effects) [65, 66]. Results from this analysis are presented in Table 3.

Recall from our summary statistics in Table 1 that women earned significantly lower hourly wages than their male counterparts, $33.03 to men's $38.50. That is a difference of $5.47 per hour. As noted previously, women working in computer science earned 86.6 cents for every dollar that a man in computer science makes; this is consistent with the 0.144 log point gender wage gap shown at the bottom of Table 3. Over a third of this gap (0.052 log points, or 36.4 percent) is explained by differences in the composition of men and women who work in computer science. Results from Tables 1 and 2 indicate both that men are more likely to be married and parents than women, and these characteristics bring wage premiums. Table 3 reveals that if women were as likely as men to be married and have children, the gender wage gap would be 0.008 log points smaller. This impact is driven almost entirely by marriage rather than children (results not shown). Yet the largest contributor to the explained portion of the gender wage gap is age. Despite no significant gender difference in mean age, men are more clustered in mid-career age groups, while women are more concentrated in early and late age groups. This could be an artifact of the retention of early female trailblazers in computer science, as well as recent encouragement focused on increasing women's presence in computer science, or a result of women being more likely to leave computer science occupations, permanently or temporarily, mid-career to accommodate family demands. Regardless of the cause, if women had the same age distribution as men, the gender wage gap would be 0.035 log points, or 24.1% smaller.

What degree field and race and ethnicity contribute to the explained variance is also qualitatively interesting. The share contributed to the explained variance in the wage gap by degree field is almost totally driven by having a non-STEM other degree (results not shown). Women are significantly more likely than men to have a non-STEM other degree (36 percent vs. men's 21 percent). If women were more likely to have a STEM or Business degree, as their male

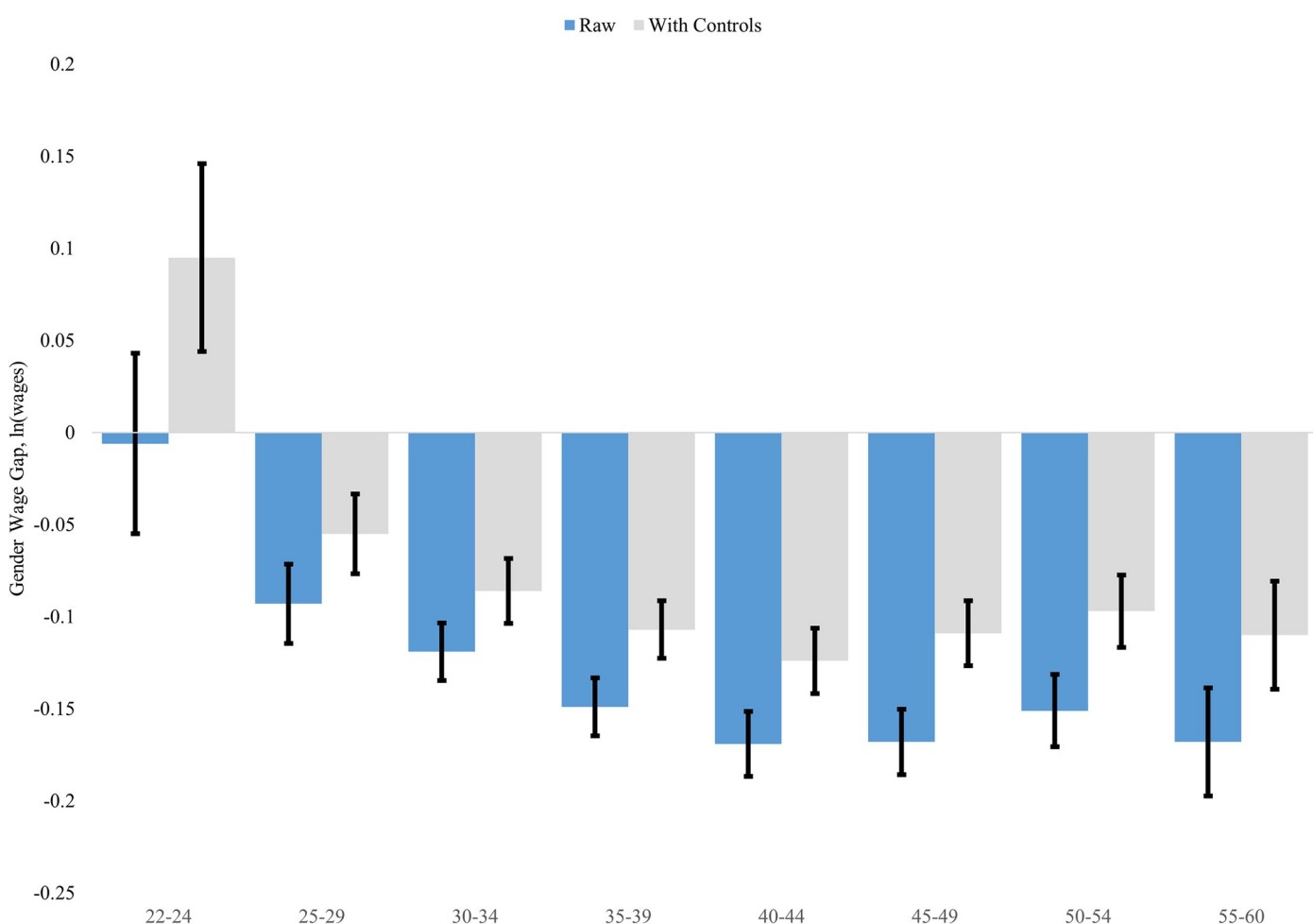

**Fig 3. Gender wage gap by age group, without and with controls.** Notes: Data come from the 2009–2019 American Community Survey. The sample is restricted to those ages 22 to 60 with a Bachelor's degree working full time (35+ hours/week) with positive income in a computer science occupation. A computer science occupation is defined by the Census Bureau in Landivar [61]. We additionally exclude those whose youngest residential child is over 18. Respondents must also have a valid birthyear and received their degree after 1980. All models are weighted using ACS person weights. Blue bars show the coefficient for Female from a regression restricted to members of that age group with no controls. The gray bars show the coefficient for Female from a regression restricted to members of the same age group with all the controls in Model 4 of Table 2. The black bars show the 95 percent confidence interval.

counterparts do, the gender wage gap would decline by 0.017 log points. The greater racial diversity among women computer science professionals further exacerbates the gender wage gap; ten percent of female computer scientists are Black, for example, versus only 6 percent of male computer scientists. Our OLS results (Table 2) revealed that Black computer scientists earned significantly less than their white counterparts. Black women's larger representation among female computer science workers therefore lowers women's average wages.

Yet nearly two-thirds of the gender wage gap (63.6 percent) is due to the unexplained component–often used as a proxy for discrimination [2, 42, 46]. Women receive different returns than their male counterparts even when their characteristics are the same. The biggest contributor to the unexplained portion of the gender wage gap is the differential return to family characteristics. If women received the same returns as men, the gender wage gap would be 6.3 percentage points lower. Like the explained portion of the gender wage gap, this family contribution is also driven by returns to marriage. If women received the same marriage premium as men (without changing their likelihood of being married), the gender wage gap would reduce

**Table 3. Decomposition of components of gender difference in log wages.**

| Effect of gender gap in explanatory variables | Explained | | Unexplained | |
|---|---|---|---|---|
| | log points | % of GWG | log points | % of GWG |
| Family | 0.008 | 5.7% | 0.049 | 33.7% |
| Age | 0.035 | 24.1% | -0.011 | -7.8% |
| Degree Field | 0.002 | 1.4% | 0.006 | 4.4% |
| Highest Degree | 0.000 | 0.0% | 0.000 | -0.3% |
| Occupation | 0.001 | 1.0% | -0.013 | -8.7% |
| Race/Nativity | 0.005 | 3.5% | -0.003 | -2.2% |
| Constant | | | 0.064 | 44.6% |
| Total | 0.052 | 36.4% | 0.092 | 63.6% |
| Total Gender Wage Gap | 0.144 | | | |

Notes: Data come from the 2009–2019 American Community Survey. The sample is restricted to full-time (35+ hrs/ week) workers age 22–60 in the field of computer science earning positive income and who earned at least a Bachelor's degree after 1979. We additionally exclude those whose youngest residential child is over 18. Entries are the male-female differentials in the indicated variables. Family includes dummy variables for Married, Previously Married, Only Child(ren) 0–4, Young and School-aged Children, and Only child(ren) 5–18 (ref = never married, childless). Age includes age and age-squared. Degree field includes dummy variables for Business degree, Engineering degree, Other STEM degree, and Other non-STEM bachelor's degree (ref = Computer Science degree). Highest Degree includes dummy variables for Professional Degree, Masters, and Doctorate (ref = Bachelor's degree). Occupation includes dummy variables for CIS Manager, Computer Scientist, Computer Analyst, Information Security Analyst, Computer Programmer, Web Developer, Computer Support Specialist, Database Administrator, Network Administrator, Network Architect, and Other CS (ref = Software Developer). Race/ Nativity includes dummy variables for Black, Asian, Other Race, Hispanic, Foreign born (ref = native born NH-White).

by 0.049 log points, almost 5 percent. The second largest contributor to the unexplained portion of the gender wage gap comes from measures of occupation, though accounting for these variables *widens* the gender wage gap. Were women to get the same returns to occupation as men, the gender wage gap would be 1.3 percentage points *larger*. This is because women are less likely to be in the highest paid occupations of Software Developer and Network Architect (though they are more likely to be Managers) and much more likely to be lower paid Computer Analysts and Web Developers, so even paying them the same as the men in their same occupation would not close the average gender wage gap in computer science as a whole as they are concentrated in lower paying occupations.

*Assessing gender differences in returns*: *Interaction effects*. Our OLS regression results from Table 2 showed that there was both a marriage and parenthood bonus; married computer science workers, as well as those who had children, earned more than their unmarried and childless counterparts. But do men and women receive similar returns to marital and parental status, or do women experience penalties for family characteristics that advantage men? To investigate this, we next present results that interacted gender by our measures of marital and parental status for our OLS results (Table 4). The first column presents the raw gender wage gap from Model 1 of Table 2. Column 2 shows the gender wage gap with controls (Model 4 from Table 2), and the fully interacted results are shown in Column 3. Our interactions reveal that marriage provides wage premiums for both women and men. Married men earn 11.1 percent more than do single men, while married women earn 9.6 percent more than do single women who work in computer science jobs. Yet because men continue to earn more than women, even single men earn significantly more–on the order of 8.6 cents per hour more– than married women, and this difference is statistically significant. Divorced men also appear

**Table 4. Interactions of family characteristics.**

| | (1) | | (2) | | (3) | |
|---|---|---|---|---|---|---|
| | Model 1 ln(wages) | | Model 4 ln(wages) | | Model 4 + Interactions ln(wages) | |
| | β | SE | β | SE | β | SE |
| Female | -0.144*** | (0.004) | -0.095*** | (0.004) | -0.013* | (0.007) |
| **Demographic Controls** | | | | | | |
| Married | | | | | 0.105*** | (0.006) |
| Prev. Married | | | | | 0.015* | (0.009) |
| Only Child(ren) 0–4 | | | | | 0.068*** | (0.006) |
| Both Young & School-age | | | | | 0.063*** | (0.007) |
| Only Child(ren) 5–18 | | | | | 0.046*** | (0.005) |
| **Interactions** | | | | | | |
| Female*Married | | | | | -0.092*** | (0.010) |
| Female*Prev. Married | | | | | -0.048*** | (0.014) |
| Female*Only Child(ren) 0–4 | | | | | -0.047*** | (0.013) |
| Fem * Both Young & School-age | | | | | -0.036** | (0.014) |
| Female*Only Child(ren) 5–18 | | | | | -0.054*** | (0.009) |
| All Controls | | | X | | X | |
| Constant | 3.480*** | (0.002) | 1.241*** | (0.036) | 1.218*** | (0.036) |
| Observations | 206,640 | | 206,640 | | 206,640 | |
| AIC | 403,528 | | 351,676 | | 351,273 | |
| BIC | 403,549 | | 352,003 | | 351,651 | |
| R-squared | 0.010 | | 0.23 | | 0.231 | |

Notes: Data come from the 2009–2019 American Community Survey. The sample is restricted to those ages 22 to 60 with a Bachelor's degree working full time (35 + hours/week) with positive income in a computer science occupation. A computer science occupation is defined by the Census Bureau in Landivar [61]. We exclude those whose youngest child in the household is over 18. Respondents must also have a valid birthyear and received their degree after 1980. All models are weighted using ACS person weights. Robust standard errors are shown in parentheses. AIC = Akaike information criterion. BIC = Bayesian information criterion.

*** p<0.001

** p<0.01

* p<0.05.

to benefit from having been married in the past, earning a marginally significant 1.5 percent more than never married men. Divorced women, in contrast, do not earn appreciably more than never married women, and earn significantly less than single men– 4.5 cents less per hour. Women are also twice as likely to be previously married than men, further reducing the share of women working in computer science who benefit from receiving a marriage premium.

Of note is that among those working in computer science occupations, both men and women with parenting responsibilities earn more than their childless counterparts. This pattern holds regardless of whether their children are pre-school aged, school aged, or a combination of the two. This parenthood premium is greatest when children are the youngest; men with only pre-school aged children earn 7 cents more per hour than do childless men, while women with pre-school aged children earn 2.1 cents more per hour than do women who are childless. The premium is somewhat smaller when they have both pre-school and school-aged children, perhaps because they have more than one child, but they still earn significantly more than their childless counterparts– 6.5 cents more per hour for men, and 2.7 cents more per hour for women, relative to their childless counterparts. The wage premium is smallest for

those whose children are school-aged– 4.7 percent more for men and negative (-1 percent) for women, relative to their respective counterparts with no children. While this difference seems small, women with any pre-school aged children (who account for 15 percent of all female computer science workers, compared with 20 percent of the men) earn significantly more than childless men; mothers of pre-schoolers earn about 1 cent more per hour, and those with both pre-school and school-aged children earn 1.4 cents more per hour than do childless men. On the other hand, while men who are fathers of school-aged children earn significantly more than do childless men, women who are mothers of school-aged children earn significantly less–about 2.1 percent lower hourly wages–than childless men–though they earn more than childless women.

How differential returns to parenthood contribute to the unexplained component of the gender wage gap can be more easily understood by graphing the returns mothers and fathers receive at different family configurations, compared to their childless counterparts. Fig 4 graphs the returns to different aged child(ren) among married parents. Relative to never married and childless men, married fathers receive a sizable parenthood premium regardless of whether their child(ren) are pre-schoolers or school-aged, across all three groups of children.

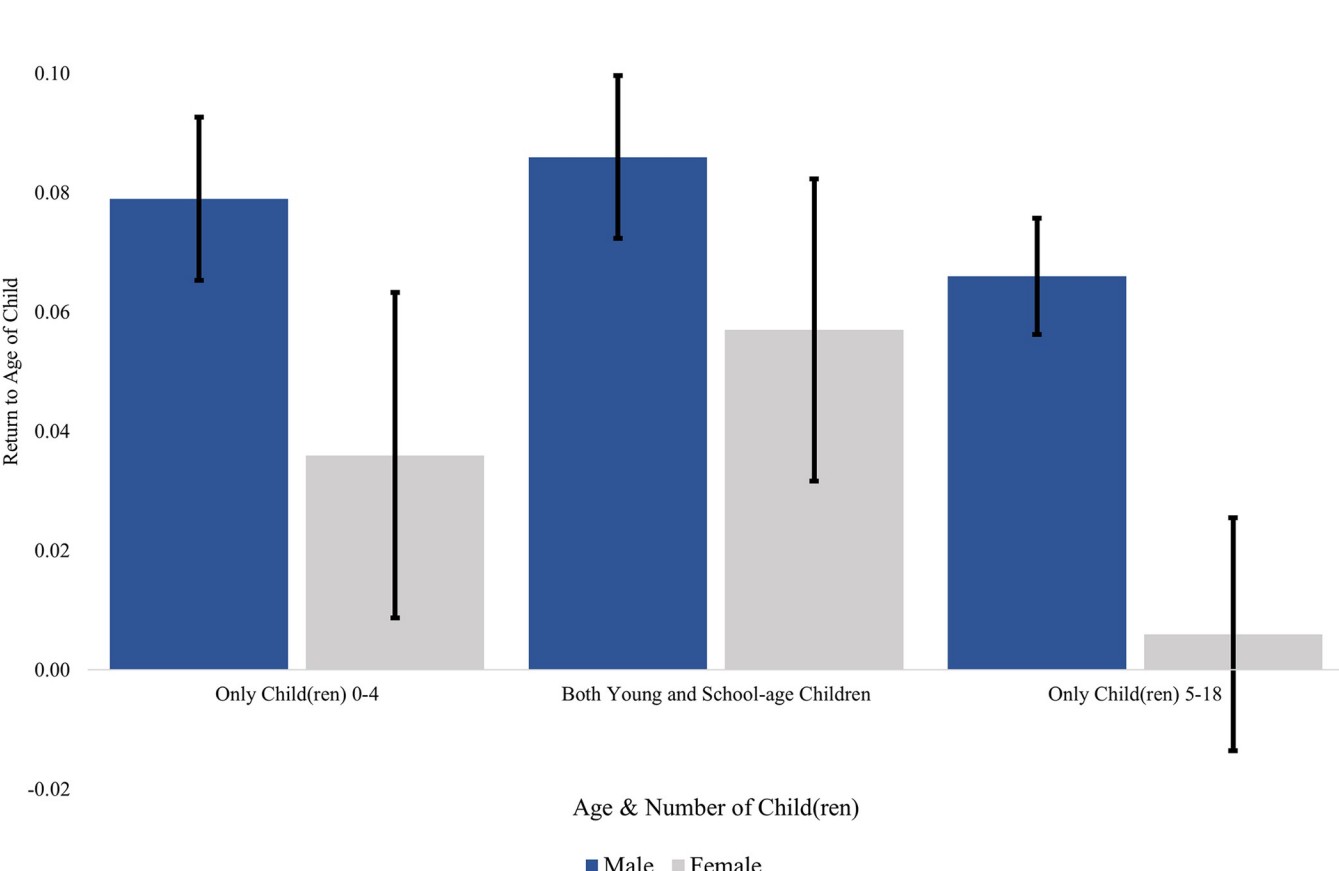

**Fig 4. Wage premiums by parent sex and children's age/number for married parents.** Notes: Data come from the 2009–2019 American Community Survey. The sample is restricted to those ages 22 to 60 with a Bachelor's degree working full time (35+ hours/week) with positive income in a computer science occupation. A computer science occupation is defined by the Census Bureau in Landivar [61]. Respondents must also have a valid birth year and received their degree after 1980. All models are weighted using ACS person weights. Blue bars show the coefficient for the designated child variable from a regression restricted to married men. The gray bars show the coefficient the designated child variable from a regression restricted to married women. All models include all controls from Model 4 of Table 2. The black bars show the 95 percent confidence interval.

Married mothers of pre-schoolers also earn a parenthood premium over their childless counterparts. The bars are all above the line (representing never married computer science professionals of the same gender), and the error bars are also above the line. But married mothers of school-aged children do not earn a significantly greater amount than do childless women. They do not earn less–but unlike men, they do not earn more than their childless counterparts. Those with only school-aged children account for a substantial proportion (28 percent) of all mothers; such differential rewards widen the unexplained component of the gender wage gap. We do not observe a parity gradient for mothers; results (not shown) indicate that mothers with two or more children earn more, not less, than those with just one child. Still, the parenthood premium remains larger for fathers than mothers, and these premium increase when men have more than one child.

How much of the persistent gender wage gap is explained by gender differences in returns to human capital attributes? Results for our gender differentiated analysis of the returns to degree field are presented in Table 5. Model 1 depicts the raw gender wage gap, while model 2 shows the gap after including all controls (copied from Model 1 and Model 4 from Table 2). Model 3 shows the interaction between female and degree field when computer science serves as the reference group. While the gender wage gap among women who graduated with college degrees in computer science is smaller than the gap for all workers in the computer science

**Table 5. Degree field interactions.**

|  | (1) | | (2) | | (3) | |
| --- | --- | --- | --- | --- | --- | --- |
|  | Model 1<br>ln(wages) | | Model 4<br>ln(wages) | | Model 4<br>+ Interactions<br>ln(wages) | |
|  | β | SE | β | SE | B | SE |
| Female | -0.144*** | (0.004) | -0.095*** | (0.004) | -0.090*** | (0.007) |
| **Degree Field** |  |  |  |  |  |  |
| Bachelor's in Engineering |  |  |  |  | 0.055*** | (0.005) |
| Bachelor's in Other STEM field |  |  |  |  | -0.028*** | (0.007) |
| Bachelor's in Business |  |  |  |  | -0.047*** | (0.005) |
| Bachelor's in Other non-STEM |  |  |  |  | -0.126*** | (0.005) |
| **Interactions** |  |  |  |  |  |  |
| Female*Engin.Degree |  |  |  |  | -0.045*** | (0.014) |
| Female*Other STEM Degree |  |  |  |  | -0.008 | (0.013) |
| Female*Bus.Degree |  |  |  |  | -0.015 | (0.010) |
| Female*Other non-STEM |  |  |  |  | 0.014 | (0.010) |
| All Controls |  |  | X | | X | |
| Constant | 3.480*** | (0.002) | 1.241*** | (0.036) | 1.240*** | (0.036) |
| Observations | 206,640 | | 206,640 | | 206,640 | |
| AIC | 403,528 | | 351,676 | | 351,646 | |
| BIC | 403,549 | | 352,003 | | 352,014 | |
| R-squared | 0.010 | | 0.230 | | 0.230 | |

Notes: Data come from the 2009–2019 American Community Survey. The sample is restricted to those ages 22 to 60 with a Bachelor's degree working full time (35 + hours/week) with positive income in a computer science occupation. A computer science occupation is defined by the Census Bureau in Landivar [61]. We exclude those whose youngest child in the household is over 18. Respondents must also have a valid birthyear and received their degree after 1980. All models are weighted using ACS person weights. Robust standard errors are shown in parentheses. AIC = Akaike information criterion. BIC = Bayesian information criterion.

*** p<0.001

** p<0.01

* p<0.05.

field, women who obtained a bachelor's degree in computer science still earn significantly lower hourly wages than their male counterparts with computer science degrees (0.090 log points, or 8.6 percent less). Furthermore, women with college degrees in engineering do not receive the same premium when they work in computer science as men with engineering degrees. Whereas men with engineering degrees earn 5.7 percent more than men with degrees in computer science, women with engineering degrees earn only 1 percent more than do women with computer science degrees and 7.7 percent less than men with degrees in computer science. In other words, women are not reaping the same rewards to their engineering degrees that their male counterparts do, with regards to hourly earnings, when they work in computer science. Men who obtained their degrees in Other STEM fields, Business, or Other Non-STEM fields experience a wage penalty when they work in computer science occupations relative to men with degrees in computer science, and women continue to earn less because of the gender penalty ($B$ = -0.090). But women's earnings disadvantage for those with college degrees in business or another non-STEM field are not significantly greater than that experienced by their male counterparts.

Table 6 focuses on interactions between gender and occupation. Consistent with the decomposition findings in Table 3, we see that the returns to being a female software developer are significantly lower than for male software developers; women software developers earn 14.8 percent less than their male counterparts. Because software developers are the highest paid occupation in computer science, all other occupations earn less, as was shown in Table 2. All interaction terms between occupation and female are positive, indicating that the gender wage gap is largest among software developers. Even though only 11 percent of women who work in computer science are in software developer jobs, this disparity helps drive the overall gender wage gap. Still, women earn less than men in all computer science occupations. Even were women to work in computer science occupations that pay the most, the gender wage gap would persist.

## Discussion

Over the past few decades, women have made great strides in pursuing education and obtaining employment in many fields that were once the purview of men, and rates of maternal employment are high [67]. Such changes have narrowed the gender wage gap, though progress in reducing earnings disparities has stalled. Increasing women's representation in STEM occupations is often promoted as one means of reducing the gap in men's and women's wages, as is improving women's attachment to the labor market. In this article we assessed to what extent the gender wage gap in computer science is attributable to family factors or human capital investments. We rely on data from the American Community Survey and utilize statistical techniques that allow us to ascertain the size of the gender wage gap and to decompose the influence of family factors and human capital measures.

Women who work full-time in computer science occupations experience a sizable and significant gender wage gap, making 13.4 cents less per dollar than their male counterparts. Controlling for differences in human capital and family characteristics narrows this gap, but women working in computer science continue to earn significantly less (9 cents per hour) than men. Compositional differences explain less than forty percent of the gender wage gap. Most of the gender wage gap among those working in computer science jobs was the result of women receiving different returns to their characteristics–the "unexplained" portion of the wage gap representing both unobserved factors as well as discrimination. We find the biggest contributors of the unexplained portion of the gap come from family and occupation. While women in our sample receive a premium for being married and having children, it is

**Table 6. Occupation interactions.**

| | (1) | | (2) | | (3) | |
|---|---|---|---|---|---|---|
| | Model 1 ln(wages) | | Model 4 ln(wages) | | Model 4 + Interactions ln(wages) | |
| | β | SE | β | SE | β | SE |
| Female | -0.144*** | (0.004) | -0.095*** | (0.004) | -0.161*** | (0.011) |
| **Occupation** | | | | | | |
| CIS Manager | | | | | -0.066*** | (0.006) |
| Computer Scientist | | | | | -0.205*** | (0.013) |
| Computer Analyst | | | | | -0.210*** | (0.007) |
| Information Security Analyst | | | | | -0.104*** | (0.013) |
| Computer Programmers | | | | | -0.225*** | (0.007) |
| Web Developer | | | | | -0.368*** | (0.012) |
| Computer Support Specialist | | | | | -0.434*** | (0.008) |
| Database Administrator | | | | | -0.236*** | (0.012) |
| Network Administrator | | | | | -0.310*** | (0.008) |
| Network Architect | | | | | -0.129*** | (0.010) |
| All Other CS | | | | | -0.304*** | (0.007) |
| **Interactions** | | | | | | |
| Female*CISManager | | | | | 0.061*** | (0.014) |
| Female*Comp. Scientist | | | | | 0.083*** | (0.024) |
| Female*Analyst | | | | | 0.065*** | (0.014) |
| Female*Info Analyst | | | | | 0.056* | (0.032) |
| Female*Programmer | | | | | 0.052*** | (0.015) |
| Female*Web Developer | | | | | 0.080*** | (0.021) |
| Female*Support | | | | | 0.095*** | (0.016) |
| Female*Data Admin | | | | | 0.037* | (0.020) |
| Female*Network Admin | | | | | 0.110*** | (0.018) |
| Female*Network Arch. | | | | | 0.097** | (0.038) |
| Female*Other | | | | | 0.110*** | (0.016) |
| All Controls | | | X | | X | |
| Constant | 3.480*** | (0.002) | 1.241*** | (0.036) | 1.254*** | (0.036) |
| Observations | 206,640 | | 206,640 | | 206,640 | |
| AIC | 403,528 | | 351,676 | | 351,578 | |
| BIC | 403,549 | | 352,003 | | 352,018 | |
| R-squared | 0.010 | | 0.230 | | 0.230 | |

Notes: Data come from the 2009–2019 American Community Survey. The sample is restricted to those ages 22 to 60 with a Bachelor's degree working full time (35 + hours/week) with positive income in a computer science occupation. A computer science occupation is defined by the Census Bureau in Landivar [61]. We exclude those whose youngest child in the household is over 18. Respondents must also have a valid birthyear and received their degree after 1980. All models are weighted using ACS person weights. Robust standard errors are shown in parentheses. AIC = Akaike information criterion. BIC = Bayesian information criterion.

*** $p < 0.001$

** $p < 0.01$

* $p < 0.05$.

significantly smaller than the premium received by men. Differential returns to family attributes (marriage, parenthood) reveals discrimination that negatively affect women and accumulates over the work life [16, 21, 57]; in fact, our evidence suggests it begins in the late twenties, before most college-educated women have become mothers [44].

Our results also point to important differential returns to degree field and occupation, further substantiating recent research highlighting occupational sorting as one mechanism contributing to inequality among highly educated workers [13, 42, 56]. Increasing the shares of women receiving bachelor's degrees in the fields with the highest returns–engineering and computer science—should reduce the gender wage gap. But the wage gap would still persist, as these are also the fields where the gender wage gap is largest. Furthermore, there is some evidence that as the proportion of women entering into previously male-dominated occupations increases, these occupations are devalued and pay less [68], including in computer science [3]. Renewed attempts to reduce the gender wage gap must go beyond encouraging more women to obtain degrees in STEM fields or to pursue jobs in programming-intensive occupations (the girls in STEM or girls who code mantra), to addressing why women earn so much less than their male counterparts in computer science, including paying attention to discriminatory practices in hiring and promotion [13, 42, 58].

There are, however, some positive signs suggesting that important headway has been made in addressing some forms of gender inequality. Several recent studies have found that the association between female caregiving responsibilities and lower returns has weakened, if not reversed [3, 6, 18, 19], and our findings provide further evidence of this for women working in computer science. First, we find evidence of a marriage premium for women working in computer science, who have earnings that are, on average, significantly higher than their never married or divorced counterparts. Married women continue to earn significantly less than married men. But that is due to men's substantially larger marriage premium, rather than resulting from a penalty that married women experience. Second, motherhood is not inevitably associated with wage penalties. Mothers continue to earn less than fathers, and the presence of school-aged children seems to be uniquely detrimental to maternal earnings. Mothers with school-aged children are also older and may have entered the work force at a time with larger gender wage penalties. Future research is needed to ascertain whether younger cohorts continue to experience a motherhood penalty for older children when they go through this stage, as we are unable to discern whether this is a cohort effect. Family roles, as wives or mothers, need not be detrimental to wages, at least among the selective group of women working in computer science jobs. That said, our results also provide hints that mothers of school-aged children continue to experience unique challenges, perhaps due to the inadequacies of school coverage, which is neither full-day nor year-round, that affect their ability to work and thus their earnings.

Of course, our analysis is not without shortcomings. We are unable to determine whether pay disparities are the result of "glass ceiling" effects, where women earn less than men with similar levels of experience as they advance in their careers. The structure of the ACS data does not enable us to ascertain when respondents began working in computer science, or their career progression. Nor can we establish if women remained in the labor force following childbirth–though the premiums they receive to parenthood strongly suggest that they do. Women who remain working in computer science occupations following parenthood may be selectively different from those who leave for other fields that may provide more flexibility–though research suggests that such jobs are often not very flexible or family friendly [48, 53]. Unfortunately, data sets that include such information as well as an adequate sample size of those working in computer science and information on work experience or a longitudinal structure are lacking. While we cannot observe firm size, tenure, prestige, or whether the firm is a start-up, the growing body of research on discriminatory practices experienced by women in computer science suggests that such factors alone are inadequate to rule out gender discrimination as a source of persistent wage inequities in computer science. Our evidence suggests that both supply and demand side factors contribute to the persistent gender wage gap.

Despite these shortcomings, our results present evidence that narrowing wage disparities between women and men remains a challenge. Even among those women who obtain degrees and work in male-dominant occupations (such as computer science), wage disparities persist. In conjunction with journalistic evidence of the chilly climate facing many women employed in computer science occupations [8–10], it is perhaps not surprising that attrition from computer science jobs is significantly greater among women than men [24]. Dissatisfaction with pay and promotion, after all, are a key contributor to exits from the STEM labor force [23]. But we also provide some evidence that in some ways, the work world–or at least jobs in computer science–have become more amenable to women seeking to be both workers and members of families. Our findings challenge the narrative that women's career opportunities are always detrimentally affected by family roles. Addressing underrepresentation in STEM fields and gender pay disparities requires tackling the ways discrimination plays out in the contemporary labor force among professional workers.

## Supporting information

**S1 File. Descriptive statistics by occupation and gender.** Notes: Data come from the 2009–2019 American Community survey. The sample is restricted to those ages 22 to 60 with a Bachelor's degree working full time (35+ hours/week) with positive income in a computer science occupation. A computer science occupation is defined by the Census Bureau in Landivar (2013). We exclude those whose youngest child in the household is over 18. Respondents must also have a valid birthyear and received their degree after 1980. Averages are weighted using the ACS person weights.
(DOCX)

## Acknowledgments

Earlier versions of this paper were presented at the 2023 annual meeting of the Population Association of America, as well as to the Institute for Advanced Study at the University of Minnesota Law School (2019). Any opinions, findings, and conclusions or recommendations expressed in this material are those of the authors and do not necessarily reflect the views of the National Science Foundation or the Federal Trade Commission.

## Author Contributions

**Conceptualization:** Sharon Sassler.

**Data curation:** Pamela Meyerhofer.

**Formal analysis:** Pamela Meyerhofer.

**Funding acquisition:** Sharon Sassler.

**Investigation:** Sharon Sassler, Pamela Meyerhofer.

**Methodology:** Sharon Sassler, Pamela Meyerhofer.

**Project administration:** Sharon Sassler.

**Resources:** Sharon Sassler.

**Supervision:** Sharon Sassler.

**Visualization:** Pamela Meyerhofer.

**Writing – original draft:** Sharon Sassler.

**Writing – review & editing:** Sharon Sassler.

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
