## [Decision Letter · Decision Letter 0]

4 Jun 2023

PONE-D-23-13914Factors shaping the gender wage gap among college educated computer science workersPLOS ONE

Dear Dr. Sassler,

Thank you for submitting your manuscript to PLOS ONE. After careful consideration, we feel that it has merit but does not fully meet PLOS ONE’s publication criteria as it currently stands. Therefore, we invite you to submit a revised version of the manuscript that addresses the points raised during the review process.

We received the reviews for you manuscript ID PONE-D-23-13914 "Factors shaping the gender wage gap among college educated computer science workers". Both reviewers were impressed with the paper and see its importance to the study of gender differences in the labor market. However, both reviewers raise concerns, especially regarding the reporting of the results. I suggest that you carefully consider the comments of the reviewers, add descriptive analysis tables and provide the additional information the reviewers asked for.  

We look forward to receiving your revised manuscript.

Kind regards,

Eyal Bar-Haim

Academic Editor

PLOS ONE

Journal Requirements:

Reviewers' comments:

Reviewer's Responses to Questions

**Comments to the Author**

1. Is the manuscript technically sound, and do the data support the conclusions?

Reviewer #1: Yes

Reviewer #2: Yes

2. Has the statistical analysis been performed appropriately and rigorously? 

Reviewer #1: Yes

Reviewer #2: Yes

3. Have the authors made all data underlying the findings in their manuscript fully available?

Reviewer #1: Yes

Reviewer #2: Yes

4. Is the manuscript presented in an intelligible fashion and written in standard English?

Reviewer #1: Yes

Reviewer #2: Yes

5. Review Comments to the Author

Reviewer #1: Review on PONE-D-23-13914: "Factors shaping the gender wage gap among college educated computer science

workers"

The paper uses data from the ACS to assess gender wage gaps among computer science workers and to examine the factors underlying the gender wage gap within this occupation. Considering the field of study, different occupations within the computer science workers, and family characteristics, the authors suggest the sources of the gender gaps.

The paper is well-written, easy to follow, the empirical approach is solid, and it is appropriate for the journal's readership. However, I have a few minor suggestions or questions that addressing them will contribute to strengthening the (already strong) paper.

1. I suggest addressing in more detail the decision to include full-time employees only. Is it because there are only a few computer scientists who work part-time? Are there gender gaps among computer science employees in work hours? Women with children often work part-time or fewer hours a week because childbearing and household responsibilities are gendered. Do the results remain the same as part-time workers are included?

2. Field of study definitions: Which fields are in the 'other' category? What are the frequencies? Other fields of study, such as mathematics and statistics, sometimes lead to positions in computer science and the decision to include computer science, engineering, and business as distinct categories and to aggregate all other fields into the 'other' category, especially since 47% of women are in the 'other' fields of study category suggests a specific explanation and justification. For example, why not use 'other STEM' and 'other'?

3. In order to achieve a better understanding of different types of occupations within computer science, I suggest adding descriptive data at the occupational level. For example – what is the average wage in each sub-occupation? Understanding the average wage (and not only the breakdown by gender) will assist the reader in understanding the rank of different types of occupations. More data at the occupational level that would be helpful will be the share of women in each type of sub-occupation, the average wage, average working hours, etc.

4. At the top row on p. 27 (row 509), you mark that "Increasing the share of women who receive bachelor's degree in these fields, than, would elevate women's wages". I think this sentence is misleading. The entrance of women into occupations is not always a proxy for increasing wages in the occupations but the other way around (for example, as was suggested by the devaluation theory[1]). I think you should address it in your discussion or results.

[1] Levanon, A., England, P., & Allison, P. (2009). Occupational feminization and pay: Assessing causal dynamics using 1950–2000 US census data. Social forces, 88(2), 865-891.

Reviewer #2: General feedback

This is an interesting, clearly written paper on a highly important topic that links the gender wage gap to a rapidly changing part of the labour market – the IT sector.

Nonetheless, I am not entirely convinced the paper is ready for publication, it has much more potential. The literature review and analysis a strong thread. The contribution, research question (and/or hypotheses) and the analytical strategy need to be laid out more explicitly. What does this article add to the large and growing body of literature? Moreover, the link between literature, objective, research question and model selection/order must be derived more convincing to make it easier to follow the numerous results (that may be somewhat overwhelming). The conclusion does not go much beyond summary of results (and data limitations).

Major points

a) Methods.

I miss descriptive statistics that allow placing the results of computer scientist workers into perspective to the overall population (similar to table 1) and in particular how to evaluate the selection “bias”. In what respect do computer scientists differ in each gender from the rest of the workers (not only earnings but all variables taken into account in the analysis education, and field of education but also proclivity to work, degree of working hours, distribution of occupations)? This may give important insights about the generalisability of the results to other populations (entire labour force or specific sectors/occupations). For instance, could we generalise the results to skilled white-collar jobs in general? Why not selecting this group including scientists, highly specialised technicians, engineers, etc.?

More specifically, here we have several steps of selection: (1) into work, (2) into full time work, (3) into computer scientist professions, as well as other selection processes into a particular job/firm. Some of these might be worth modelling. So perhaps, even more useful could be a methodology following two steps Heckmann selection method. The authors consider this option and why have they applied the present method?

With respect to the sample, I find the lower age limit of 22 a bit odd, given the high skilled nature of the field and the inclusion of MA and PhD in the sample. Perhaps justify with similar studies, many use 25 or even 35.

I do not understand the construction of the parental status variables. The dummies do not seem to be mutually exclusive? Any children vs any children under 5 vs any children under 1? Vs number of children? There seems to be multicollinearity or the labels are not specified in line with the method? If there is a dummy for children under 5 and a dummy for children and an indicator for the number of children, what does a specific combination of these mean technically and in practice? Also for marital status I am not convinced that the variable is constructed meaningful. Why would it be relevant to know if a person was married before? Where are the theoretical arguments for including this category? Furthermore, it does not seem very relevant. I would speculate that single parent status can add more to the understanding. Please revise the construction of the family background to yield another combination of marital status and children. Use different models to distinguish between number and age of children – this will allow you to see which one is (more) relevant and perhaps drop one option to the annex as robustness check.

Also note that Joshi et al 2019 (IZA WP NO 12725) find an inverse U-shaped gender wage gap over their life-course where an initial gap in early adulthood widened substantially during childrearing years, affecting earnings in full-time and part-time jobs.

How many fixed effects are there precisely? Region does not really seem to add much? Or time? Split into two models, so see if geographical or temporal dimension adds more and drop if model does not improve. This increases the model df enormously and does not explain much additional variation.

This being said, model evaluation diagnostics are not included (AIC, BIC). The R squared is not enough/recommended, R square should be used if df changes.

If hourly wages are analysed, why are part-time workers excluded (the fact that it gives similar results is not an argument to drop these important part of the female workers but to included it!)? Much research argues that part-time work is crucial to understand the gender wage gap.

I suggest including the 2020 update nevertheless. The COVID argument does not convince me to exclude the most recent wave, at least not without a robustness test.

Why are occupations and regions clustered at this level / in these groups? How do you justify the choices? With the data more refined occupational differences could be made (refined within-occupational gender gap as potential contribution as this is a problem with many data sets).

b) Results.

What is the rationale of the model sequence? Right now, it seems somewhat arbitrary, as poorly aligned with the literature review.

There are a vast number of models. Perhaps focus on one or two important sets and also refine the literature review accordingly. This is specifically important as all results are significant using data with such large sample size. That’s also the crux: which results are (the most) relevant and meaningful? Split the paper or focus by dropping some models or place some into an annex. Model 2 and 3 can be collapsed, they are both ethnic/racial/migration background and do not make a big difference in terms of gender gap and R squared. On the contrary, the human capital model (4) yields bigger changes while 4 variable sets (with plenty of df) are introduced: does it make sense to split them up to uncover which variable contributes most to decreasing the gender gap or explaining data variation? Why do you introduce fixed effects so late? Does it make sense to keep them given their low impact? Which model is the best given parsimony principle? (e.g. the model with occupational interactions does not add explanative power but many dfs)

I am not entirely convinced of time-constant geo-fixed effects and this also contradicts your argument of rapidly changing Silicon Valley. Moreover, there are cohort effects, which should be acknowledged somewhere. The meaning of IT degrees and demand has changed tremendously over the past decade(s) and thus over birth/graduate cohorts. Several cohort studies show the gender wage gap could narrow to the extent that women were lacking behind men in terms of educational attainment. Across more recent cohorts, the gender wage gap remains flat. (Campbell et al 2013 Soc Sci Res, Bar-Haim 2022 in Social Indicators Research)

The raw gender wage/pay gap seems to be expressed incorrectly (abstract ; lin2 321 for instance): it should probably be 18%, not 82%. The GWG is defined as the difference in hourly earnings M-F divided by M hourly earnings multiplied by 100, as for instance in computer science occupations =($5.44/$38.68)*100=14% (overall mean hourly wage not given in the text).

c) Theory.

What’s the contribution of the paper? Does it add with a particular angle or new data/vairables? What is the precise research question and/or hypotheses that will structure the analysis?

I would refrain from talking about discrimination in the paper using such a research design. The unexplained gap is likely to reflect (also) many omitted variables such as refined skills and degrees, work experience, attitudes, etc.

Minor points

Line 300f: Be more precise: do you mean college degree instead of degree? People may have other degrees in computer science that are not observed in the data.

Line 69f: please describe the results of these studies in more detail (using which concept? Full-time vs part-time, period, etc.).

Please mention in the methods section whether gross or net wages used.

Please improve the visualisation of the results in the figures. Labels and scales are not visible, y axis are not labelled to understand units.

Is the size of firm significant? Labour supply is a joint household decision. Do you have information on the household level (e.g. spouse working)?

The literature is somewhat US centred. While this is not a disadvantage as the paper analyses US data, the phenomenon might have conclusions that can be generalised beyond the US.

6. PLOS authors have the option to publish the peer review history of their article (what does this mean?). If published, this will include your full peer review and any attached files.

Reviewer #1: No

Reviewer #2: No

---

## [Author Response · Author response to Decision Letter 0]

20 Aug 2023

RESPONSE TO REVIEWERS

We would like to thank the editor and reviewers for their time and thoughtful comments. We have paid close attention to the suggested revisions and below describe the changes made in response to their suggestions, as well as justifications when we did not alter the manuscript as recommended. We appreciate the helpful advice of the reviewers, and believe our paper is stronger and clearer as a result.

Our marked-up copy of the manuscript highlights the changes made to the original version. We highlighted new additions in yellow; sections that are deleted have been highlighted in blue. 

To preview the most important changes: 

1. We have reorganized and strengthened the theory section (R2) and as a result reduced the number of models we present, linking the models more closely to theory and eliminating the fixed-effects models (R2). 

2. Second, we have further clarified our field of study definitions to provide additional detail on Other STEM degrees (R1), to create an “Other STEM” and “Other Non-STEM” degree field. We have also added a (Supplementary) descriptive table at the occupation level, detailing the average wage of each sub-occupation, the proportion of women working in each field, and the mean hours worked weekly (R1). 

3. In reviewing our coding, we discovered that our sample included about 12,000 respondents who had graduated from 1970 to 1979 who were working in computer science, though our manuscript stated that respondents had to have graduated in 1980 or later. We therefore exclude these respondents. Our analysis now focuses on those who earned their college degrees since 1980. This reduces our sample size, minimizing the numbers of those whose children would be grown or who are nearing retirement age. We have changed the Ns for our sample to reflect this.

4. Coding of Children: We have addressed the construction of the parental status variables to make the categories mutually exclusive and easier to interpret, and also hypothesized the justifications for including a category for the previously married. 

5. We have addressed devaluation theory more in our discussion and further emphasized our paper’s contribution to the burgeoning literature on gender disparities in earnings and the factors perpetuating gender inequality, linking it more tightly to the literature and modeling choices. 

Below, we provide a response to each point raised by Reviewers 1 and 2, noting where our response is to both of them, or to one individually.

Reviewer 1, Point 1, and Reviewer 2: Both Reviewers ask about our decision to focus only on full-time employees, to the exclusion of part-time workers. There are several reasons why we limit our analysis to full-time workers. The first is theoretical. Our paper starts off with questioning those who assert that closing the gender wage gap will require that all workers – especially women – major in STEM fields and follow “ideal worker” norms of full-time employment. We want to focus on those who have done so, as that best enables us to explore what kind of pay-offs exist to following that script, and to ascertain whether returns to men and women who follow that route are similar. 

Our second explanation is based on established research practices. We stand on the shoulders of previous scholars who have studied the gender wage gap. In their 2017 overview of the gender wage gap, Blau and Kahn focus on full-time workers to present levels and trends in the gender wage gap. They justify their approach, saying it “is designed to identify female and male workers with fairly similar levels of labor-market commitment.” They report that sensitivity analyses which included those employed part time or part year yielded very similar results; we followed in their footsteps, and also found relatively little change in the gender wage gap when part-time workers were included. The sociological research is more prone to include part-time workers, but even there, recent studies report that the largest wage gaps were concentrated among the highest-paid highly educated workers, who tend to work long hours, rather than part-time workers (Quadlin, VanHeuvelen, Ahearn, 2023). 

Third, we provide an empirical basis for our decision. Only 4.24% of those working in computer science jobs are employed part-time. Women are disproportionately represented among part-time workers (7.6% of all female workers in CS work part time, compared with 3% of men). We note the small proportions of part-time workers on Page 11, when explaining that they have been excluded. Supplementary Table 1 shows differences between full- and part-time workers, and between part-time men and women. Women who work part-time do not differ significantly from men in the hours they put in, though their hourly wages are significantly lower. That said, part-time workers differ significantly from their full-time counterparts across most dimensions. They are significantly younger and Whiter, and far more likely to have obtained their college degree in “other non-STEM” majors, and significantly more likely to work as web developers or computer support specialists, both areas with the lowest hourly wages, than full-time workers. 

Reviewer 1 asks whether family factors shape why women in computer science work only part time. Our evidence suggests that family obligations may be a better predictor of whether men work full-time versus part-time than whether women do. There is less of a family dichotomy among women working part- or full-time than there is among men. The majority of women working in computer science are married regardless of whether they work full- or part-time – 67% of those working part-time versus 58% of those working full-time. Only 33% of men working part-time are married, compared to 66% of men working full-time. Women part-time workers are also far more likely than their part-time male counterparts to be parents (56% of PT women versus 20% of PT men), but the gap in the likelihood of being parents between part-time and full-time working women is much smaller (14% difference) than it is among men (28% difference), where 52% of men working full-time have minor children in the home. For women, then, it does not seem to be the case that family factors are the exclusive factors differentiating those working part-time from those engaged in the work force full-time, whereas they do appear to be so for men.

As one would expect, part-time workers do earn significantly less than those working full-time, but the gender gap in earnings among part-time workers is much smaller than among full-time workers; the raw gender wage gap for part-time workers is only $2.07 (vs. $5.47 among full-time workers). Including part-time workers in our analysis changes results only minimally (the coefficients for gender, showing the size of the gender wage gap, are shown in the table below; complete models are available from the authors). Results of a multivariate analyses including part-time workers shows that the overall male-female gender wage gap is .165 log points; when it is restricted to those working full-time it is .144 log points (a 15.2 cent difference, versus a 13.4 cent difference). Including all controls, the remaining gender wage gap is 0.108 log points, or 10.2 cents per dollar (compared to -0.095 log points, or 9.1 cents per dollar, for full-time works (seen in Table 2 in the submitted paper). Additionally, the model fit, as measured by the AIC and BIC, are better when we restrict our analysis to full-time workers. We therefore argue that to achieve our goal of understanding the factors shaping the gender wage gap in computer science requires focusing on full-time workers, the group where the gender wage gap is greatest. In order to clarify the purpose of this paper, we have added full-time to the title which is now “Factors Shaping the Gender Wage Gap among Full-Time, College Educated Computer Science Workers.”

Linear Regression Predicting Log Hourly Wages for Full- and Part-Time Workers

• Reviewer 1, Point 2: Reviewer 1 asked about which fields were in our “Other” category, and suggested we not group other STEM majors with all other majors. We have therefore created a separate category called “Other STEM,” and our catchall “Other non-STEM” group now consists mainly of various social science and humanities graduates. We have modified the sentence detailing the coding of field of study.

Ten percent of those working full-time in computer science have college degrees in STEM fields other than computer science or engineering. Women are slightly more likely than men to be found in this “other STEM group” (11% of women versus 10% of men. Nearly one-third of this group have degrees in Mathematics and Statistics, with over a quarter (28.7%) having their degrees in the Physical Sciences, and another fifth (21.8%) in the Life Sciences (including biology) (not shown in paper). Our regression analyses indicate that those who received their bachelor’s degree in “other STEM” earn significantly less than their counterparts with degrees in computer science or engineering, but our interaction results of degree by gender indicate that women with this degree do not earn significantly less than men.

The share in “other non-STEM degrees” is now smaller, making up only a quarter of those working in computer science jobs, though women working in computer science jobs are significantly more likely than men to have other non-STEM college degrees. The group with other non-STEM degrees is still quite large (n = 53,018), and well populated with respondents who obtained degrees in the social sciences and humanities. The five most common majors for those in this group are: 1) Social Sciences (n = 11,702); 2) Communications (n = 6,266); 3) Fine Arts (n = 5,733); 4) Psychology (n = 4,374); and 5) English Language, Literature, and Composition (n = 3,695). We thank Reviewer 1 for suggesting this additional refinement of the measure. 

Reviewer 1, Point 3: Reviewer 1 wanted more information at the occupation level (share of women, average wage, average working hours Appendix Table 1 includes the proportion of women employed as well as average wages, usual hours worked, and the sample size for all and by gender for each occupation. The proportion of women working in various computer science occupations varies, ranging from a low of 11% of all network architects (which has among the highest earnings), to 36% of all computer analysts and web developers. At the bivariate level, there is no clear association between occupations where women are underrepresented and earnings. It is also evident that while women work fewer hours, on average, than their male counterparts in every occupation, this gap is rather small, maxing out at 1.26 hours more per week for those men working as CIS managers relative to women CIS managers.

Reviewer 1, Point 4: We have modified our discussion on the expected role that increasing women’s receipt of STEM degrees should have on wages, softening that language, referencing devaluation theory, and then providing a reference to work that has found it operating in computer science. We also include references to newly published studies (as of May 2023) that also argue for the need to better understand gender differences to returns to field of study and educational attainment that take better account of discrimination (e.g., Quadlin, VanHeuvelen, & Ahearn, 2023; Zheng & Weeden, 2023) to further strengthen our assertion.

We summarize and address Reviewer 2’s suggestions in order, to the best of our ability:

Reviewer 2, Point 1: Reviewer 2 asks us to better connect the paper to the literature and motivate the order or results more clearly. We have re-ordered the literature review and findings (described in more detail) to address this.

Reviewer 2, Point 2: R2 asked us to place the results of computer scientists into perspective to the overall population, to ascertain how computer scientists may differ from the rest of workers. As much as we would love to do that, such an endeavor goes well beyond the scope of this paper and would require substantial time for data shaping and analysis. Data from the ACS is not well-suited to understand factors associated with returns to education in professions such as law, medicine, or academia, where prestige of one’s graduate degrees plays a much larger role in initial placement and pay than in computer science, where the majority of workers have only a Bachelor’s degree. Other scholars have focused on professional occupations such as medicine and law (Buchman and McDaniel, 2016; Noonan, Corcoran, and Courant, 2005), STEM fields broadly (Beutel and Schliefer, 2022; Cech & Blair-Loy, 2019), and academia (Shauman), but with other data sets that contain far more information on the prestige of graduate schools and other factors that may be associated with pay and the gender wage gap (such as GPA, course work, time to degree). Furthermore, most other professions require graduate degrees and licensing (legal careers, medicine, academia) that are not essential in computer science occupations. 

We do not include discussion about the generalizability of our findings beyond computer science because we argue that it is essential to focus on computer science, rather than STEM fields or professions more broadly, due to the unique position computer science jobs occupy within STEM. As we note in our paper, half of all STEM jobs are in computer science. Existing research finds that computer science differs in important ways from other STEM fields, including engineering. It is the one field, for example, where the share of women obtaining degrees has fallen in recent years, rather than increased (even as the proportion of women obtaining STEM degrees has risen, and the majority of all college degrees are now awarded to women). But much of the research on STEM groups computer science and engineering with other STEM fields where women’s experiences are radically different. There are other important reasons to focus only on computer science, relating to challenges with retention, promotion, and advancement into leadership. We do discuss how our findings are similar to those found for other professions in our conclusion, but ascertaining how computer scientists differ from other workers, or why they are selected into computer science professions, is outside the scope of this paper and, unfortunately, not feasible with this dataset.

Reviewer 2, Point 3: Reviewer 2 also asks about selection into work, or full-time work, as well as into computer science professions, and then into a particular job or firm. This paper, and particularly this dataset, are not well-suited to explore this. Heckman’s methods might allow us to assess whether those who work full-time are selectively different from those working part-time, but selection models best fit when there are numerous variables that are also not included in the models for the main analysis and are more commonly used with panel or event study data. We lack those with our ACS data, which has strengths in terms of large sample size, but weaknesses when it comes to detailed measures of factors shaping selection into the occupation. Additionally, with our focus on full-time workers (explained above), selection into work is more complicated. Are they selecting into any work or full-time work and what variables would predict that accurately? Cross-sectional data is particularly ill-suited for this analysis. 

We did contemplate conducting propensity score matching, to assess whether there were respondents with a greater/lesser likelihood of working full time and if that then shaped wages, but we are hampered by an absence of predictor variables in the ACS that would differ from those that also appear in our main analytic model. The ACS lacks many of the types of information that are available in surveys (like the NSLY, or the Survey of Doctorate Recipients), such as work aspirations, course work taken in college, math assessments or grade point average, gender ideology, adolescent expectations for parenthood, or other indicators that are useful to get at selection. Furthermore, the proportion of both women and men who are working part-time is quite small – as we already discussed (see Supplementary Table 1). The authors have published other papers that explore transitions into computer science occupations with other data sets; that is not the goal of this paper. This paper seeks to understand what factors contribute to the gender wage gap among those working full-time in computer science, and to what extent this gap is explained by composition or returns. 

Reviewer 2, Point 4: Reviewer 2 asked us to explain our sample age restrictions, suggesting that we raise the age limit to 25 or 35, given the skilled nature of the field. Of note is that computer science is a field that does not require an advanced degree, and in fact is characterized by relatively low proportions with advanced degrees (see Table 1). It is not necessary to have additional schooling to get a well-paid job in CS. Furthermore, there have been reports that among recent graduates, the gender gap is not significant or that women earn more – making it important to incorporate the youngest computer science workers in our analysis. We also find this, shown in the new Figure 3. We reran our analyses for CS workers aged 25 to 60. While the raw gender wage gap was similar, in the final model the gender wage gap for those 25 to 60 was slightly larger (as we would hypothesize, given the high salaries offered to those with the most recent degrees and, ostensibly, in-demand skills); women ages 25 to 60 made 9.6 cents less per hour than men did, whereas among women 22 to 60, women made 9.1 cents less per hour than their male counterparts. The model fit is better in terms of more explained variance (R-Squared) in the more expansive model (including those aged 22 to 60). We therefore continue to utilize the original age range. 

Reviewer 2, Point 5: Construction of parental status variables: We agree with R2 that these measures can be confusing. To allay Reviewer 2’s concerns, we have reconstructed our child measures. We create three, mutually exclusive, dummy variables to capture the differential care demands by age and number of children. The first dummy equals 1 if all the children in the household are less than 5 years old (this may be only one child or multiple children). The second dummy equals one if the respondent has at least one child under 5 and at least one child ages 5-18 (school-aged). The third dummy equals one if all the children in the household are school-aged (5-18). We feel this measure best captures the burdens parents face based on the age and number of children in the household while being mutually exclusive. 

Additionally, Reviewer 2 asked why, theoretically, we distinguished between currently married, previously married, and never married. We now include a sentence in our review of the literature on marriage to denote that those who were previously married but divorced might experience a marriage premium, but that following a divorce negative effects are often found and generally affect women more strongly than men (page 7). Descriptive results from Table 1 indicate that women working in CS jobs are twice as likely as their male counterparts to be previously married (12% of all women versus 6% of all men). We therefore distinguish between currently and previously married and also interact gender and previously married to reveal that the impact of being previously married also differs for men and women. Men receive a slight premium from marriage even when it has ended, but that is not the case for women, who experience a wage penalty for being previously married (though they still earn more than never married women). Given that 12 percent of women working in computer science are divorced, we feel this is an important distinction to keep. While it would be interesting to explore the impact of single parent status on the wage gap, the small sample size prevents us from doing that. 

Reviewer 2, Point 6: With regards to our age effects, we are unable to explore the gender wage gap over the life-course as our data are cross-sectional, and we lack even measures of work experience. To address this concern to the best of our ability within the constraints of the data, we show and discuss how the gender wage gap evolves across the age groups studied (ages 22 to 60) in the new Figure 3 and reference the Joshi (2021) paper in discussing age effects. Reviewer 2 points to cohort studies showing that the gender wage gap could narrow as women “catch up” to men in terms of educational attainment. Given the cross-sectional structure of the ACS data, we do not have a cohort study and cannot address this in this dataset. Any period or cohort measure is highly correlated with age. We do reference various studies that utilizes the SESTAT (Scientists and Engineers Statistical Data System) data; their sample size of computer scientists, however, is considerably smaller and less recent than what the ACS allows.

Reviewer 2 , Points 7-15: Reviewer 2 makes several comments about our model specification which we address in order below and reference a special OLS table demonstrating our findings:

• Region and year fixed effects: As suggested, we explored whether adding geographic or temporal measures added significantly to the model of the raw gender wage gap, running them separately and together. Columns 2-4 in the above table show that region and year FE have little impact on the gender wage gap and do little to improve model fit. Since they did not explain much additional variation and come at the cost of additional degrees of freedom, we have excluded these measures in the paper. 

• The next concern was the lack of AIC and BIC in the tables. They have been added to all relevant tables in the draft.

• R2 expressed concern about excluding part-time workers. Please see explanation above as R1 also expressed this concern and it is addressed above.

• R2 suggests including 2020. In addition to our concerns stated in the paper that we do not want to include observations during the pandemic or recovery, the Census Bureau has stated that the data for 2020 is unreliable due to collection issues that occurred during the pandemic. Consequently, we continue end our sample at 2019.

• R2 asks why we group regions and occupations at the level we do. We are no longer including regions, so we will not address that concern. While SOC codes go to the 6 digit-level, the ACS only utilizes down to the 4-digit level. Consequently, this is the finest occupation level we can observe in this dataset.

• Additionally, Reviewer 2 asked us to rethink the number of models, as well as how they are sequenced. We have done so, and reorganized them to better reflect the literature review, and now focus on the main measures of interest – the family measures, and then the human capital measures. R2 also expressed concern that so many human capital variables are added simultaneously. In the table above, Column 5 matches Table 2, Model 2 in the paper adding family and age controls. Column 6 adds degree field and highest degree controls while Column 7 adds occupation. Separating the human capital controls into Columns 6 and 7 has little impact on the gender wage gap or model fit according to the AIC and BIC. Accordingly, we combine them into Model 3 of Table 2 in the paper for the sake of parsimony. We have incorporated our final set of controls (for race-ethnicity-nativity) into Table 2, Model 4, as while important, they do not add substantially to the model fit, but are relevant when it comes to the decomposition results. We thank Reviewer 2 for highlighting the importance of parsimony. Point 15 asks about time-constant region FE and cohorts. Both have been addressed above.

Reviewer 2, Point 16: We have checked all reports of the raw gender wage gap to make sure they are expressed correctly. It is customary in the gender wage gap literature to express the gap as “cents on the dollar” or “women earn X% of what men earn”. This is why we express it as 82 percent instead of 18 percent.

Reviewer 2, Point 17 asks us to clarify the contribution of the paper, which we have done. In light of persistent arguments that mothers are disadvantaged in the labor market – particularly in the aftermath of the COVID pandemic – we seek to highlight that in fact, in a growing number of professions, mothers and wives are not experiencing wage penalties – at least relative to single, childless women. Contesting this narrative is important, as young adults making their career and relationship decisions are not immune to negative media stories, and women are particularly susceptible to stories suggesting they should forego career investments, or family. In fact, a growing number of young women express uncertainty about their abilities to balance family and work. Our research is important in highlighting that there is some good news in the battle to address gender inequality, in that we find less evidence that mothers (and wives) receive a wage penalty. That should be reassuring for those interested in pursuing professional careers and having a family. 

Minor Points (18-24):

• Reviewer 2 suggests that we refrain from discussing discrimination. Although we feel it is important to discuss this possibility, we have softened our language to ensure we acknowledge this is just one possibility and omitted variables and other causes are also reasonable explanation.

• Point 19: We have made our description of degrees more precise; here, we refer to college degrees.

• Point 20: Reviewer 2 asks for more detail on studies (69f). We discuss these results in greater detail in the section on Family Explanations.

• Point 21 inquires if we use gross or net wages. The ACS only provides gross wages. We clarify this in our definition section.

• Point 22 expresses concerns about labeling figures. We acknowledge the labels are small but that is only in accordance with the journal’s submission requirements for figures. We have added Y-axis labels.

• Point 23, Size of firm: We are unable to assess whether the size of firm is significant; such information is not available in the ACS data. We also focus on the individual level data, rather than including information about whether partners (spouses or cohabiting partners) are employed. Such information is available in the CPS but not the ACS. We do reference other research that has utilized survey data with information on spousal employment (and field of employment) (Glass, Sassler, Levitte, & Michelmore, 2013). 

• Point 24, US focus in literature: While some western countries, like the UK, have similar small shares of women working in computer science, in many other countries – especially those that are sometimes termed less developed – women are better represented in STEM fields of study and jobs (Charles, 2011). But family factors may play quite a different role, as would gender hierarchies and queues, in other cultural contexts. We therefore limit our discussion to the United States. 

Literature referenced in letter:

• Beutel AM, Schleifer C. (2022). “Family structure, gender, and wages in STEM work.” Sociological Perspectives. 64(4), 790-819. 

• Blau FD, Kahn LM. (2017). “The Gender Wage Gap: Extent, Trends, and Explanations.” Journal of Economic Literature. 55(3):789-865. 

• Buchmann C, McDaniel A. (2016). “Motherhood and the wages of women in professional occupations.” RSF: The Russell Sage Journal of the Social Sciences. 2(4):128-150. 

• Cech EA, Blair-Loy M. (2019). “The changing career trajectories of new parents in STEM.” PNAS. 116(10):4182-4187. 

• Charles, M. (2011). “What Gender Is Science?” Contexts, 10(2), 22–28. 

• Glass, J, Sassler, S, Levitte, Y, Michelmore, K. (2013). “What’s So Special about STEM? A Comparison of Women’s Retention in STEM and Professional Occupations.” Social Forces. 92(2):723-756.

• Joshi, H, Bryson, A, Wilkinson, D., Ward, K. (2021). “The gender gap in wages over the life course: Evidence from a British cohort born in 1958.” Gender, Work, and Organization. 28(1):397-415, SI.

• Noonan, MC, Corcoran, ME, Courant, PN. (2005). "Pay Differences Among the Highly Trained: Cohort Differences in the Sex Gap in Lawyers' Earnings." Social Forces. 84 (2): 853-872.

• Quadlin N, VanHeuvelen, T, Ahearn, C. (2023). “Higher education and high-wage gender inequality.” Social Science Research. 112;102873.

• Zheng H, Weeden KA. (2023). “How Gender Segregation in Higher Education Contributes to Gender Segregation in the U.S. Labor Market.” Demography. 60(3): 761-784. 

 

Table S1. Summary Statistics by Sex including Part-Time Workers.

Notes: Data come from the 2009-2019 American Community survey. The sample is restricted to those ages 22 to 60 with a Bachelor's degree with positive income in a computer science occupation. A computer science occupation is defined by the Census Bureau in Landivar (2013). Respondents must also have a valid birthyear and received their degree after 1980. Averages are weighted using the ACS person weights. Full-time is defined as working 35+ hours/week on average.

---

## [Decision Letter · Decision Letter 1]

10 Oct 2023

Factors shaping the gender wage gap among college educated computer science workers

PONE-D-23-13914R1

Dear Dr. Sassler,

We’re pleased to inform you that your manuscript has been judged scientifically suitable for publication and will be formally accepted for publication once it meets all outstanding technical requirements.

Kind regards,

Eyal Bar-Haim

Academic Editor

PLOS ONE

Additional Editor Comments (optional):

Reviewers' comments:

Reviewer's Responses to Questions

**Comments to the Author**

1. If the authors have adequately addressed your comments raised in a previous round of review and you feel that this manuscript is now acceptable for publication, you may indicate that here to bypass the “Comments to the Author” section, enter your conflict of interest statement in the “Confidential to Editor” section, and submit your "Accept" recommendation.

Reviewer #1: All comments have been addressed

Reviewer #2: All comments have been addressed

2. Is the manuscript technically sound, and do the data support the conclusions?

Reviewer #1: Yes

Reviewer #2: Yes

3. Has the statistical analysis been performed appropriately and rigorously? 

Reviewer #1: Yes

Reviewer #2: Yes

4. Have the authors made all data underlying the findings in their manuscript fully available?

Reviewer #1: Yes

Reviewer #2: Yes

5. Is the manuscript presented in an intelligible fashion and written in standard English?

Reviewer #1: Yes

Reviewer #2: Yes

6. Review Comments to the Author

Reviewer #1: As I wrote in the first round - this paper is well-written and easy to follow, and the empirical approach is solid. In general, the paper is appropriate for the journal's readership. It focuses on the gender wage gap within the field of computer science, which contributes to understanding the gender wage gap in general and the specific patterns of employment and wages among highly skilled and highly paid men and women.

The authors have addressed my concerns, and I look forward to seeing this paper in print.

Reviewer #2: The authors have addressed all reviewers' comments, either by implementing them thoroughly into the paper or by making a convincing argument. It is now clearer to understand data limits and why they have not taken certain methodological choices.

It is an captivating paper on an highly relevant topic, I look forward to seeing it published in PLOS ONE.

However, one sentence is ambiguous in the new version, as for analysing cohort change one does not necessarily need *panel* data, see Bar Haim et al 2023. This study shows that the role of education for explaining gender earnings gaps is limited and has declined over time - while gender gaps esp. unexplained differences persist.

Consider rephrasing:

484 Additional research is necessary to ascertain if there has been cohort change in the wage gap at

485 similar ages, though that would require panel data with information on earnings across the life

486 course for several cohorts.

Reference:

Bar-Haim, E., Chauvel, L., Gornick, J.C. et al. The Persistence of the Gender Earnings Gap: Cohort Trends and the Role of Education in Twelve Countries. Soc Indic Res 165, 821–841 (2023). https://doi.org/10.1007/s11205-022-03029-x

7. PLOS authors have the option to publish the peer review history of their article (what does this mean?). If published, this will include your full peer review and any attached files.

Reviewer #1: No

Reviewer #2: No

---

## [Editor Report · Acceptance letter]

20 Oct 2023

PONE-D-23-13914R1 

Factors Shaping the Gender Wage Gap among College-Educated Computer Science Workers 

Dear Dr. Sassler:

I'm pleased to inform you that your manuscript has been deemed suitable for publication in PLOS ONE. Congratulations! Your manuscript is now with our production department. 

Kind regards, 

on behalf of

Dr. Eyal Bar-Haim 

Academic Editor

PLOS ONE